# Expanded catalog of microbial genes and metagenome-assembled genomes from the pig gut microbiome

Congying Chen[1,2,3 ✉], Yunyan Zhou[1,2], Hao Fu[1], Xinwei Xiong[1], Shaoming Fang[1], Hui Jiang[1], Jinyuan Wu[1], Hui Yang[1], Jun Gao[1] & Lusheng Huang [1,3 ✉]

Gut microbiota plays an important role in pig health and production. Still, availability of sequenced genomes and functional information for most pig gut microbes remains limited. Here we perform a landscape survey of the swine gut microbiome, spanning extensive sample sources by deep metagenomic sequencing resulting in an expanded gene catalog named pig integrated gene catalog (PIGC), containing 17,237,052 complete genes clustered at 90% protein identity from 787 gut metagenomes, of which 28% are unknown proteins. Using binning analysis, 6339 metagenome-assembled genomes (MAGs) were obtained, which were clustered to 2673 species-level genome bins (SGBs), among which 86% (2309) SGBs are unknown based on current databases. Using the present gene catalog and MAGs, we identified several strain-level differences between the gut microbiome of wild boars and commercial Duroc pigs. PIGC and MAGs provide expanded resources for swine gut microbiome-related research.

---

[1] State Key Laboratory of Pig Genetic Improvement and Production Technology, Jiangxi Agricultural University, Nanchang 330045, China. [2] These authors contributed equally: Congying Chen, Yunyan Zhou. [3] These authors jointly supervised this work: Congying Chen, Lusheng Huang. ✉email: chcy75@hotmail. com; Lushenghuang@hotmail.com

Domesticated pigs provide the majority of meat for human consumption and also serve as an animal model for biomedical research studies[1]. The gastrointestinal tract of swine harbors trillions of bacteria, which play vital roles in host metabolism, immunity, and even behaviors[2–4]. Several studies have reported an association between gut microbiota and pig feed efficiency[5,6], growth[7], and diarrhea resistance in early-weaned piglets[8]. Most of these studies relied on the available annotation information of microbes that are often connected to partial genome sequences. There remains a large portion of microbial genes that lack functional annotations or have no hit in the current database[9]. Reference genes as well as high-quality microbial genomes are essential resources for understanding the functional role of specific microbes and quantifying their abundance in the gut microbiome[10]. However, ~40–50% of gut microbial species lack reference genomes[11].

Compared with the 16S rRNA gene sequencing that is subject to bias, low sensitivity, and the lack of functional information on the gut microbiome[12,13], metagenomic sequencing can be used to infer the biological functions of microbial communities and has been gradually used to test the association between the gut microbiome and host phenotypes and diseases via metagenome-wide association study[14–16]. In fact, bottlenecks are often encountered when using existing bioinformatics tools for metagenomic sequencing data due to the need for extensive computational support. Misassembles and chimeric contigs can be created and introduce significant biases into the results[17,18]. Assembly-free-based metagenomic approaches enable to profile low-abundance organisms that are insufficient to assemble due to low sequence coverage[19], and can fast determine functional capacity of gut microbiome by aligning metagenomic sequence reads to a microbial reference gene catalog[20]. However, it is still difficult to profile those uncharacterized microbes using the assembly-free approaches. The deficiency of both annotated genes and reference genomes severely restricts the mining and use of metagenomic sequencing data and thus presents a major challenge for metagenomic sequencing analysis. Therefore, the constructions of both a complete gene catalog and a complete genomic catalog are required urgently for the studies of gut microbiome.

To date, reference gene catalogs of the gut microbiome have been reported for the human[9,21,22], dog[23], monkey[24], mouse[25,26], rat[27], and chicken[28]. A reference gene catalog of the pig gut microbiome has also been constructed using 287 fecal samples, containing 7.7 million nonredundant genes (named PGC in this study)[29]. However, the sequencing depth used in the construction of these catalogs was relatively low (3.31–7.0 Gb/sample). In addition, only fecal samples from domestic pigs were used. As for the genome catalog of the gut microbiome, several studies have reconstructed large numbers of microbial genomes from metagenomic sequencing data in both human[9,30–32] and agricultural animal species[33,34] using metagenomic assembly approach. However, metagenome-assembled genomes (MAGs) of gut microbiota have rarely been applied and reported for pigs[35]. A comprehensive genomic catalog of uncultivated microbiota is useful to the studies of gut microbiome although their corresponding microbial entities of MAGs need to be further confirmed by culture-based approaches.

In this work, we construct an integrated gene catalog and recover MAGs of the pig gut microbiome by sequencing five hundred samples from a wide range of sample sources spanning various ages, sexes, breeds, geographical areas, domestication, and gut locations. Especially, the lumen samples from jejunum, ileal, and cecum are used to improve the representation of this integrated gene catalog on the microbiome of the whole intestinal tract. Furthermore, the dataset of the PGC catalog[29] is also integrated into the construction of the catalog. We show gene catalogs (named pig integrated gene catalog, PIGC) of the pig gut microbiome consisting of 48,697,887 (PIGC100), 17,237,052 (PIGC90), and 7,246,447 (PIGC50) nonredundant genes at 100%, 90%, and 50% amino acid identity, respectively. In addition, a total of 6339 MAGs are recovered, which are clustered to 2673 species-level genome bins (SGBs), of which more than 86% (2309) have no available genome sequence in the current database (unknown SGBs, uSGBs). To demonstrate the value of these resources, we use the catalogs of microbial genes and MAGs to compare the gut microbiomes between wild boars and commercial Duroc pigs, which represent pigs raised in two distinctly different conditions (free-living vs. standard farm-raised in the pig industry) to identify the detailed microbiome differences between these two cohorts.

## Results

**Description of samples and metagenomic sequencing data**. The metagenomic sequencing data from 787 samples were used in this study, including 500 samples sequenced in this study. The 500 samples sequenced in this study included 472 feces, 20 cecum lumen, 6 ileal lumen, and 2 jejunum lumen contents from 8 different breeds of pig or Western × Chinese cross populations from eight farms. The pigs varied in sex and age and were raised under different feeding management conditions (Supplementary Table 1). High-throughput sequencing of DNA samples generated 5.73 Terabases (Tb) of high-quality clean data from 500 samples and achieved an average sequencing depth of 11.46 Gb/sample (Supplementary Table 1). Gene catalog data from the 287 pig gut metagenomes reported previously[29] were downloaded from NCBI and are included in this study.

**Establishment, and assessment of the quality and representation of pig gut microbiome gene catalogs**. The workflow of data processing is shown in Supplementary Fig. 1. After de novo assembly, gene prediction, integration of the previously reported gene catalog[29], and filtration of incomplete genes, 126,545,050 complete genes were identified. These genes were clustered at the protein level following the model of UniRef[36] at 100%, 90%, and 50% amino acid identity to form PIGC100, PIGC90, and PIGC50, respectively. After further filtering out those genes belonging to eukaryotes (except fungi) in each gene catalog, 48,697,887 (PIGC100), 17,237,052 (PIGC90), and 7,246,447 (PIGC50) protein clusters were generated (Fig. 1, Supplementary Fig. 1). The cluster number of PIGC90 was significantly lower compared with PIGC100 (~65%), but the numbers of known proteins in the catalog and annotated taxa were not reduced as much (only 11.0 and 3.0%) (Supplementary Fig. 2). Therefore, PIGC90 was used for further comparison and annotation analysis.

Rarefaction analysis suggested that the number of PIGC90 clusters approached a saturation point when the sample number reached 100, which is in line with previous estimates[29] (Supplementary Fig. 3). The number of protein clusters in the PIGC90 was six-fold compared with PGC, which contained 3,460,040 complete genes from a total of 7,685,872 nonredundant genes from 287 pigs, and 2,847,252 complete protein clusters at 90% protein identity (defined as PGC90)[29]. To assess the representation of this gene catalog beyond the study cohorts, five pig gut metagenomic datasets were downloaded from the public database and mapped against PIGC90. Better ratios of mapped sequence reads were obtained with PIGC90 (ranging from 87.03 to 97.83%) compared with PGC90 (ranging from 54.65 to 88.72%) (Supplementary Fig. 4). Furthermore, based on Uniprot TrEMBL, 4,818,537 (28%) clusters in the PIGC90 are unknown proteins (Supplementary Fig. 5a). These results imply

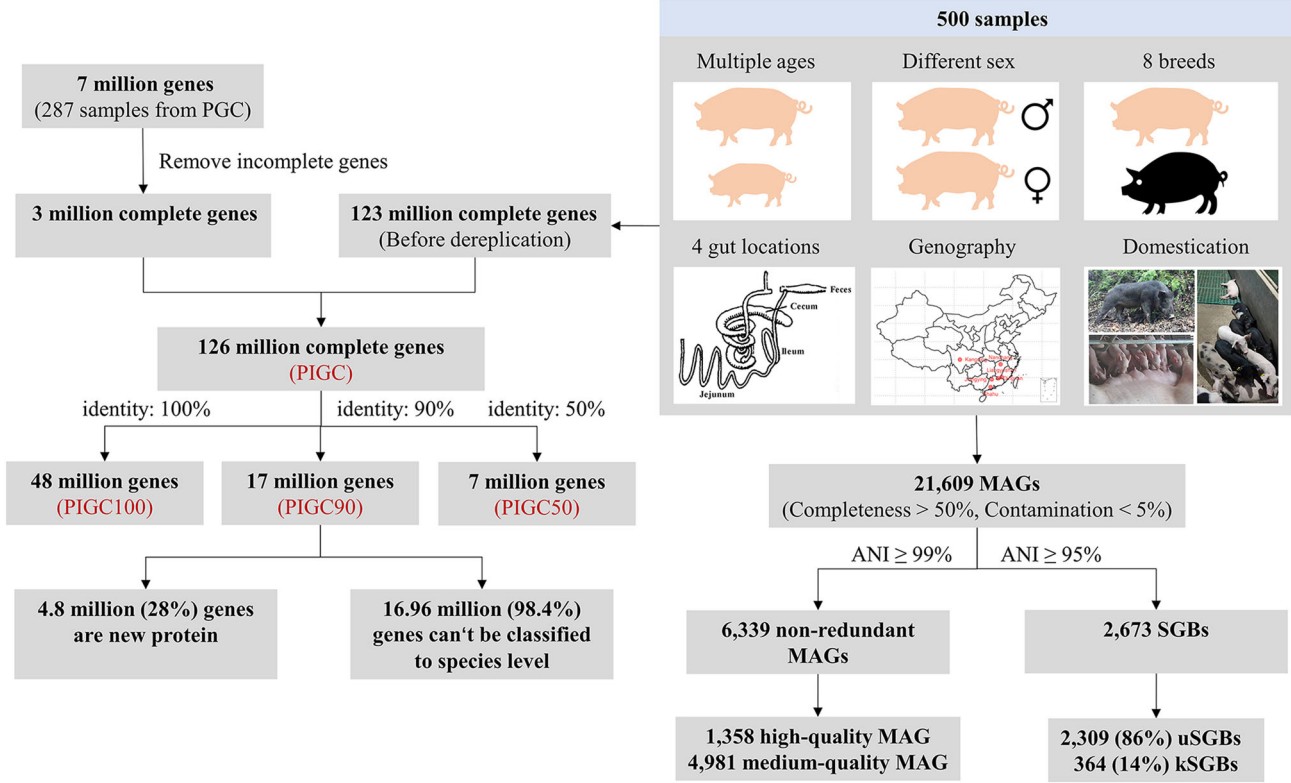

**Fig. 1 Pipeline for the construction of pig integrated gene catalog (PIGC) and metagenome-assembled genomes (MAGs).** Metagenomic sequencing data from the samples spanning age, sex, breed, gut location, geography, and domestication, as well as a pig gene catalog (PGC) from 287 metagenome data were integrated and used to construct the PIGC catalog. The complete genes were clustered at 100, 90, and 50% amino acid identity to generate nonredundant gene catalogs of PIGC100, PIGC90, and PIGC50. The reconstructed microbial genomes were clustered to strain-level and species-level genome bins (SGBs) at 99% and 95% of the average nucleotide identity (ANI), respectively. The 6339 nonredundant MAGs were divided into medium-quality MAGs (more than 50% completeness and <5% contamination) and high-quality MAGs (more than 90% completeness and <5% contamination). SGBs containing at least one reference genome (or metagenome-assembled genome) in the Genome Taxonomy Database (GTDB) were considered as known SGBs (kSGB). The SGBs without reference genomes were considered as unknown SGBs (uSGBs).

that, compared with the previous database, PIGC significantly extended the gene number of the pig gut microbiome.

**Contribution of sequencing depth and extensive sample sources to gene content of the PIGC.** We first evaluated the contribution of sequencing depth to the capture of gut microbial genes. Correlation analysis in 301 feces samples from $F_6$ pigs of the Mosaic population revealed a steady increase in the number of genes identified following the sequencing depths ($P < 2.2 \times 10^{-16}$) (Fig. 2a). The cut-off analysis of different sequencing depth in 20 samples with high sequencing depth (12.4 Gb) further suggested a significant influence of sequencing depth in the capture of microbial genes ($P = 1.7 \times 10^{-7}$; Supplementary Fig. 6). However, the distribution of unique gene numbers following gene abundances, and the percentages of the genes shared among samples indicates that most of the 17,237,052 genes were at lower abundance/prevalence within the individual samples (Fig. 2b, c).

Extensive sample sources from different ages, breeds, gut locations, and geographical conditions, especially the samples from wild boars, allowed us to evaluate the contribution of sample sources to the gene number and representation of the PIGC90. Among the 17,237,052 protein clusters (nonredundant genes), 2,843,245 genes were sample source-specific (16.5%). Feces-specific genes from adult domestic pigs occupied most of the sample source-specific genes (94.0%) likely due to the large sample size ($n = 427$). Samples from piglets contributed 78,565 specific genes. With the exception of the feces samples,

samples from different gut locations (small intestine and cecum lumen) contributed 169,125 nonredundant genes (including small intestine lumen-specific, cecum lumen-specific, and small intestine and cecum lumen-shared genes in domestic pigs and wild boars) (Fig. 2d). To our knowledge, this is the first study to include the cecum lumen and feces samples from wild boars for constructing the gene catalog of pig gut microbiome. These samples provided 95,302 wild boar-specific nonredundant genes (Fig. 2d). We further analyzed the abundances of these sample source-specific genes in the corresponding sample source that they came from. Notably, high proportions of small intestine lumen-specific (65.3%), piglet sample-specific (39.7%), and wild boar sample-specific genes had the abundances of ≥ average abundance of gene set. However, most of the feces sample-specific genes of domestic pigs (97.4%) showed low abundances in the samples that they came from (Fig. 2e). This result suggested that the utilization of the samples from different gut locations and wild boars provided very useful gene set to improve the representation of the PIGC catalog.

**Taxonomic and functional characteristics of pig gut microbiome based on the PIGC90.** Of the 17,237,052 nonredundant genes in PIGC90, 12,418,515 can be blasted to the Uniprot TrEMBL (known proteins) (Supplementary Fig. 5a). Among these, only 1,745,932 genes could be taxonomically classified (Supplementary Fig. 5b). More than 98.9% of the classified genes were assigned to bacteria, whereas the remaining 1.1% belonged

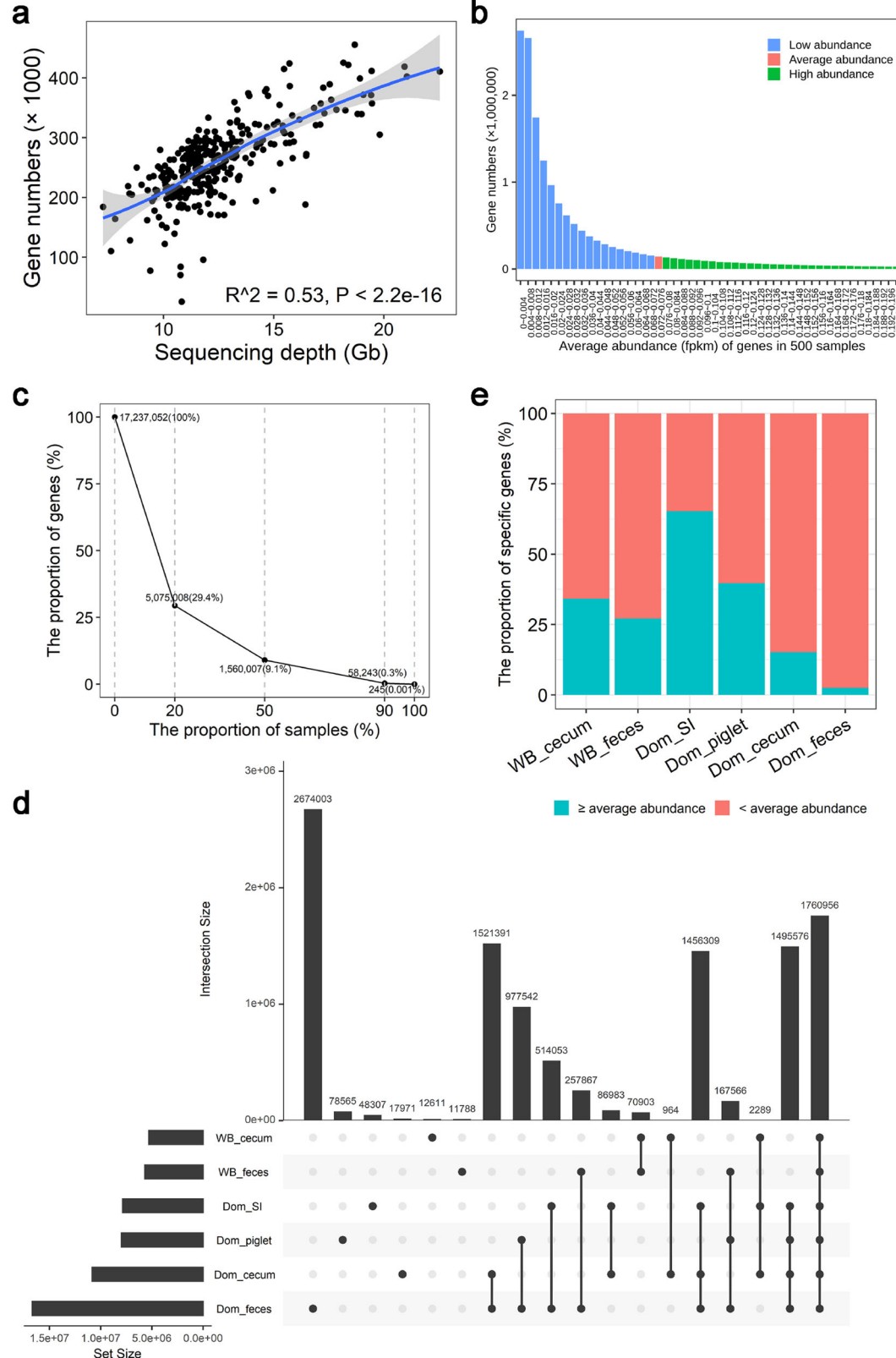

to viruses and archaea (Supplementary Fig. 5c). These classified genes were annotated to 38 phyla, 705 genera, and 1280 species. Of the 38 phyla, Firmicutes (65.5%), Bacteroidetes (14.0%), Proteobacteria (10.1%), and Actinobacteria (7.1%) were predominant (Supplementary Fig. 5d). The distribution of annotated bacterial taxa in the 500 samples was further analyzed. The

bacterial taxa detected in more than 90% of the tested samples were defined as the core bacteria of pig gut microbiome and 19 (50% in all annotated phyla) phyla, 234 (33%) genera, and 254 (20%) species were identified as core bacteria. Among these, 15 phyla, 135 genera, and 97 species were detected in all 500 samples (Fig. 3a). The abundances of these 97 species occupied more than

**Fig. 2 Contribution of sequencing depth and sample sources to the gene content of the PIGC. a** Association of predicted gene number with the sequencing depth ($n = 301$). The predicted gene number was increased significantly following the sequencing depth. Adjusted R-squared and $P$-values were calculated by a linear regression model in R (v3.6.2). **b** The distribution of the gene numbers following the relative abundances. The gene abundances shown in the x-axis were average abundance of each gene (fpkm) in 500 samples. The blue bars indicate the numbers of genes under the average abundance, the green bars indicate the numbers of genes above the average abundance, and the red bar shows the number of genes at the average abundance. **c** The numbers (percentages) of nonredundant genes in the PIGC90 shared among different proportions of samples. The values next to the dot indicate the number and proportion of genes in the PIGC90 shared among each proportion of samples. Most of the genes are low prevalence.
**d** Contribution of different sample sources to gene content of the PIGC. All 500 samples were divided into six subsets, including feces samples from wild boars (WB_feces, $n = 6$), lumen samples from the cecum of wild boars (WB_cecum, $n = 8$), the feces samples from adult domestic pigs (Dom_feces, $n = 427$), lumen samples from the cecum of adult domestic pigs (Dom_cecum, $n = 12$), lumen samples from the small intestine of adult domestic pigs (SI, $n = 8$), and feces samples from piglets (piglet, $n = 39$). Vertical bars represent the number of genes shared between the specific study sets highlighted with black dots in the lower panel. Horizontal bars in the lower panel indicate the total number of genes contained in each sample subset. **e** The proportions of sample source-specific genes having high abundance ($\geq$ average abundance) in the corresponding samples that they came from.

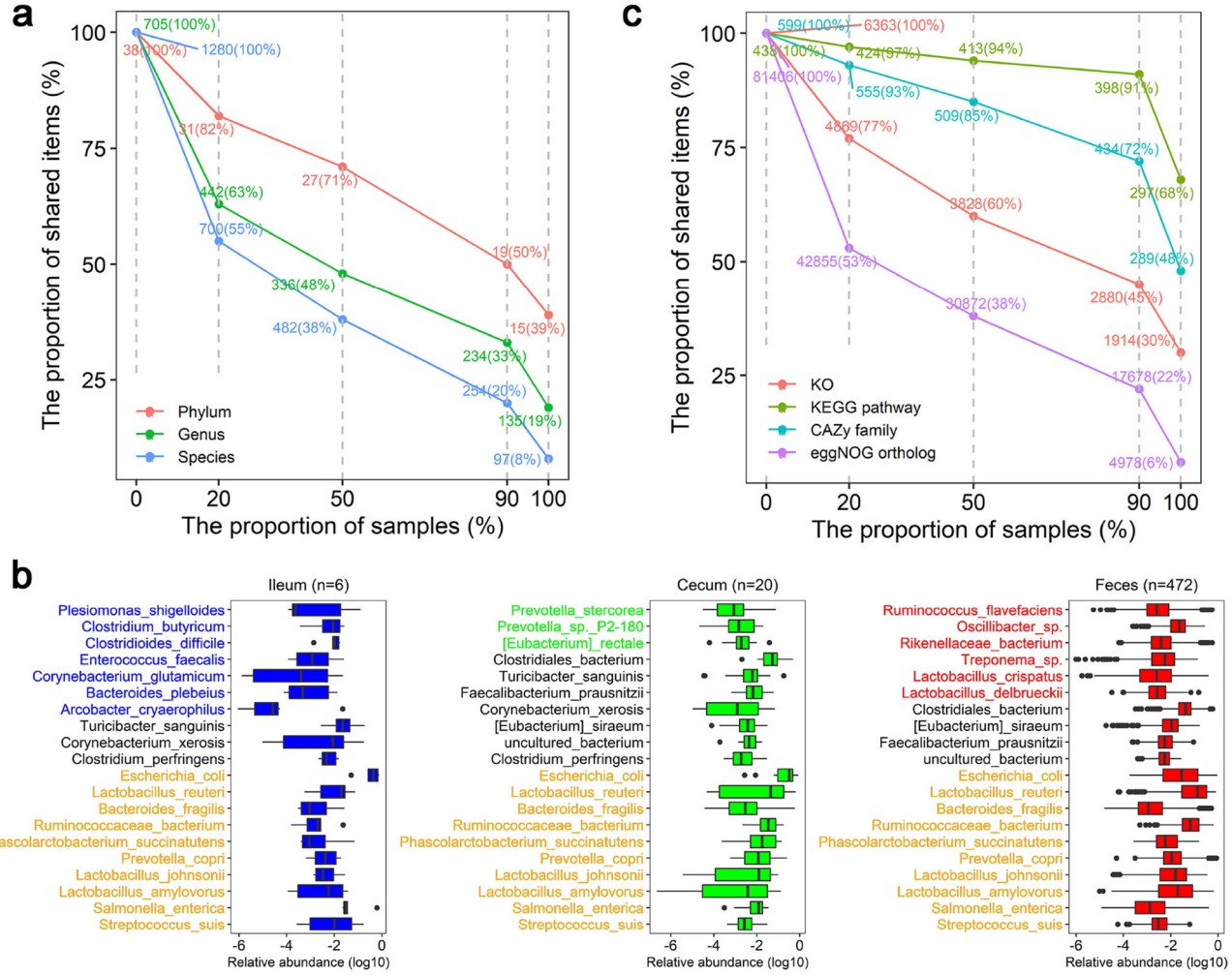

**Fig. 3 Core bacterial taxa and functional capacities of pig gut microbiome. a** Numbers (percentages) of shared bacterial taxa among different proportions of samples at the phylum (red), genus (green), and species (blue) level. The percentage of shared items and the proportion of shared samples are represented on the y- and x-axis, respectively. The number and the percentage for each item that are shared in 20, 50, 90, and 100% of samples are indicated in the Figure. Nineteen phyla, 234 genera, and 254 species were shared in 90% samples and defined as core bacteria. **b** The top 20 bacterial species in relative abundances in ileum lumen, cecum lumen, and feces, respectively. The yellow color indicates the species in the top 20 lists of all three gut locations, and the colors corresponding to boxplots show the top 20 species specific to each gut location. The $\log_{10}$ (relative abundance) values are shown on the x-axis. **c** Numbers (percentages) of shared function items among different proportions of samples for KEGG orthologues (red), KEGG pathways (olive), CAZy family (cyan), and eggNOG (purple). Other legends are like (**a**). Boxplots show median, 25th and 75th percentile, the whiskers indicate the minima and maxima, and the points laying outside the whiskers of boxplots represent the outliers.

92.29% of the total abundance of the 1280 annotated bacterial species, suggesting their high prevalence and important roles in the gut microbiome of pigs. The top 20 bacterial species in abundance in feces, ileum lumen, and cecum lumen samples are listed in Fig. 3b. Ten bacterial species, including *Lactobacillus reuteri*, *Lactobacillus johnsonii*, *Lactobacillus amylovorus*, *Ruminococcaceae bacterium*, *Escherichia coli*, *Prevotella copri*, *Bacteroides fragilis*, *Streptococcus suis*, *Phascolarctobacterium succinatutens*, and *Salmonella enterica*, are in the lists of all feces, ileum, and cecum samples. However, there were 16 bacterial species whose abundance achieved the top 20 in only one gut location. This result suggests a significant difference in the relative abundances of the same bacterial species in different gut locations. For example, consistent with our previous report[37], *Clostridium* (*Clostridium butyricum* and *Clostridium perfringens*) and *Clostridioides difficile* are highly abundant in ileum lumen samples, though only three and five bacterial species were detected that are specific for ileum and cecum lumen, respectively (Supplementary Fig. 7).

Using KEGG, CAZy, and eggNOG for functional annotation of genes, 16.56% (2,853,603) of nonredundant genes in the PIGC90 could be annotated to 6363 KEGG orthologous groups (KOs) and 438 KEGG pathways; 61.54% (10,606,969) of the genes were annotated to 81,406 eggNOG orthologous groups. In addition, 11.86% (2,045,161), 0.01% (996), and 12.68% (2,184,919) of the nonredundant genes in the PIGC90, could be classified to the Carbohydrate-Active enZYmes (CAZy) families, the Comprehensive Antibiotic Research Database (CARD), and the Virulence Factors Database (VFDB), respectively (Supplementary Table 2). The predominant functional capacities of the gut microbiome in pigs, based on the PIGC90 are listed in Supplementary Table 3. Compared with the core species, there were significantly higher percentages of KOs (45%), KEGG pathways (91%), eggNOG orthologous (22%), and CAZymes (72%) found in more than 90% of the 500 samples (core functional capacities) (Fig. 3c), implying the functional redundancy of the gut microbiota. However, the utilization of the samples from different gut locations and wild boars did not significantly increase the number of KEGG pathways and CAZymes (Supplementary Fig. 7).

**Reconstruction of microbial genomes from gut metagenomic sequencing data**. Microbial genomes were constructed from the metagenomic sequencing data obtained from the 500 samples described above. High-throughput deep metagenomic sequencing generated 21,609 MAGs at a threshold of >50% completeness and contamination of ≤5%. These reconstructed microbial genomes were then compiled and dereplicated at 99% of the average nucleotide identity (ANI). A final set of 6339 nonredundant MAGs were obtained (Supplementary Data 1). Among these, 4981 MAGs satisfied the medium-quality criteria (more than 50% completeness and <5% contamination), and 1358 MAGs showed high-quality (more than 90% completeness and <5% contamination) (Supplementary Fig. 8a)[38,39]. Most of the high-quality MAGs had a contig number <250 and ≥18 of the standard tRNAs (Supplementary Fig. 8b, c). Of the 1358 high-quality MAGs, 36 had the 5S, 16S, and 23S rRNA genes together with at least 18 tRNAs, and conformed to the MIMAG standards for the 'high quality' MAG set by the Genomic Standards Consortium[39]. All MAGs showed high prevalence in the 500 metagenomes; for example, 5211 MAGs were identified in all 500 samples, and the other MAGs (1128) existed in at least 92.6% of the 500 metagenomes (Supplementary Data 1), suggesting that these MAGs belong to strains of the core species.

The 6339 MAGs were subsequently classified into taxa using the Genome Taxonomy Database Toolkit (GTDB-Tk). All 6339

MAGs were classified to the kingdom level (6285 to bacteria and 54 to archaea), the vast majority of MAGs (6219, 98%) were assigned to the family level, 4783 (75%) were assigned to the genus level, and only 865 MAGs were assigned to 365 known species (Fig. 4a). Some representative genomes of the MAGs could not be matched to any isolated genomes in the current database, suggesting the representation of potential new species.

The 6339 MAGs were further organized into species-level genome bins (SGBs) at an ANI threshold of 95%. This clustering analysis resulted in a total of 2673 prokaryotic species, of which 2309 (86.38%) SGBs represented species without any publicly available genomes and were defined as unknown SGBs (uSGB). The taxonomic context of SGBs was obtained by recursive clustering of SGB representatives at the phylum-level genetic divergence. More than 69.1% (1846) of SGBs belonged to Firmicutes and 312 to Bacteroidetes (Fig. 4b, c). The 2309 uSGB were widely distributed in different phyla. The percentage of uSGBs of the total SBGs of each phylum are shown in Fig. 4c. Furthermore, about 73.55% of the uSGBs contained only one reconstructed genome, representing relatively rare pig-associated microbes; however, of the top 10 SGBs with the largest number of reconstructed genomes, eight were uSGBs (Supplementary Data 2).

**Implementation of the PIGC and MAGs: comparison of gut microbiome between wild boars and highly selected commercial Duroc pigs**. Free-living wild boars scavenge grass, roots, and fruits as foods for survival. Comparatively, Duroc is one of the most commonly used commercial pig breeds which are raised in the uniform farms of the modern pig industry living in high density in captivity and fed formula feeds that contain high levels of protein and energy. Whether there are distinct differences in gut microbial composition and functional capacity between wild boars and Duroc pigs is largely unknown. In this study, the PIGC was used to compare the gut microbiome between free-living wild boars and Duroc pigs from two farms. The farm-raised Duroc pigs were given high energy and protein formula feed (see "Methods"). Distinct gut microbial compositions were observed among the wild boars, Duroc-SH, and Duroc-JY (Supplementary Fig. 9a). Compared with Duroc pigs, wild boars had higher α-diversity of the gut microbiome at the genus level although the difference was not achieved significance level between wild boars and Duroc-JY (Supplementary Fig. 9b). However, this difference was not observed at the species level (Supplementary Fig. 9c). This should be caused by the poor annotations of metagenomic sequencing data at the species level. A significantly lower α-diversity was observed in Duroc-SH pigs, which might be caused by the extremely high abundance of *P. copri* in the gut of Duroc-SH (the overgrowth of *P. copri*, 54.06% in relative abundance) (Supplementary Fig. 9b–d). We then focused on those bacterial species enriched in wild boars or Duroc pigs using the PIGC90. It was discovered that the gut microbiome of wild boars had a significantly higher abundance of bacterial species from *Bacteroides* (10 species) and *Bifidobacterium* (4 species), *Hungatella hathewayi*, and *Alistipes* (5 species) (Fig. 5a, Supplementary Data 3). However, the species from *Prevotella* (23 species) and *Lactobacillus* (8 species), which are associated with porcine fat accumulation and lean meat percentage were enriched in both Duroc populations (e.g., the relative abundance of *P. copri* achieved 23.38% and 54.06% in Duroc-JY and Duroc-SH, respectively) (Fig. 5a and Supplementary Fig. 9d). It is worth noting that the abundance of *Streptococcus* species, such as *Streptococcus suis*, which is one of the pathogens that influences pig production around the world, was significantly higher in Duroc pigs (Fig. 5a).

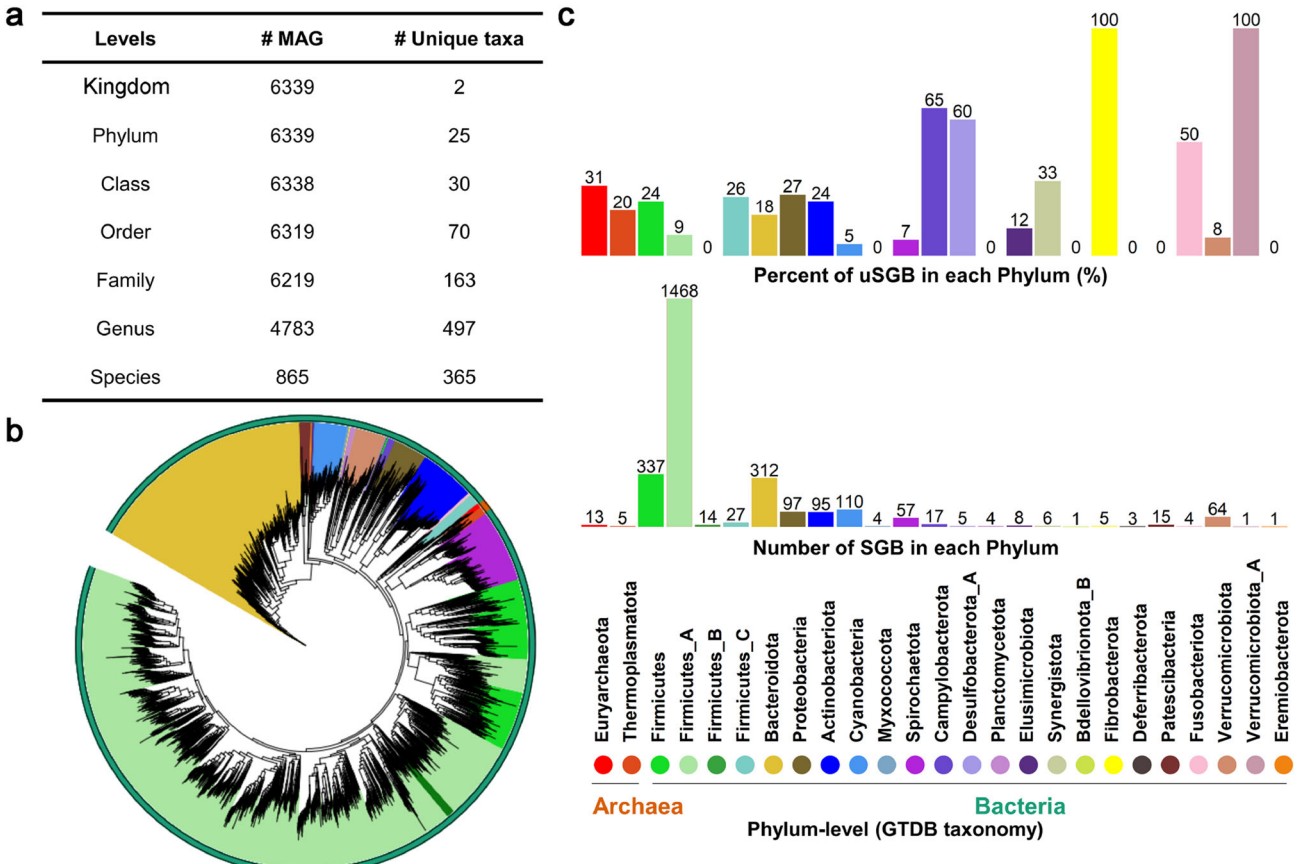

**Fig. 4 Taxonomic annotation and phylogenetic tree of 6339 metagenome-assembled genomes (MAGs). a** Taxonomic classification of 6339 MAGs at different levels. **b** Phylogenetic tree of MAGs. The outer cycle represents kingdoms and the different colors of the background of clades represent phylum. The tree was constructed by PhyloPhlAn (v3.0.51) and visualized by iTOL (v5.6.2). **c** The number of species-level genome bins (SGBs) and the percentage of unknown SGB (uSGB) in each phylum. Two phyla are from Archaea and the others belong to Bacteria. The SGBs without existing reference genome (could not be annotated at the species level by GTDB-tk) were defined as unknown SGBs (uSGBs), while the SGBs having at least one MAG could be annotated at the species level as known SGBs (kSGBs). The color of each phylum was consistent with (**b**).

MAGs were then implemented to further compare the gut microbiomes of wild boars and Duroc pigs at the strain level (MAGs) (Supplementary Data 4). Some MAGs from the same SGB showed different directions of enrichment between wild boars and Duroc pigs in a total of 7 SGBs annotated to bacteria, of which 6 SGBs were uSGBs (Fig. 6a, b). For example, among 19 MAGs that were clustered into SGB_2312 annotated to Oscillospirales, 2 MAGs were enriched in wild boars, 2 MAGs in Duroc pigs, and the other 15 MAGs showed no differential abundances between Duroc pigs and wild boars (Fig. 6a, b). Similarly, SGB_600 was a uSGB containing 33 MAGs and belonged to Methanomethylophilaceae in Archaea, and there were four and two MAGs enriched in wild boars and Duroc pigs, respectively (Fig. 6c, d). This indicates differences in the gut microbiome at the strain level between wild boars and Duroc pigs, which could not be detected without a powerful MAGs database.

The functional capacities of the gut microbiome were then compared between wild boars and Duroc pigs using the PIGC90. A total of 103 pathways showing differential abundances were identified, including 92 pathways enriched in wild boars and 11 pathways having higher abundances in Duroc pigs. The pathways enriched in the gut microbiome of wild boars were mainly related to amino acid biosynthesis and metabolism, lipid metabolism (e.g., secondary bile acid biosynthesis, fatty acid biosynthesis, and degradation), carbohydrate metabolism, vitamin metabolism (e.g., Vitamin B6 and biotin metabolism), antibiotic biosynthesis (e.g., neomycin, kanamycin, and gentamicin biosynthesis), whereas the gut microbiome of Duroc pigs was mostly enriched by the pathways associated with genetic information processing (e.g., DNA replication and homologous recombination) (Supplementary Data 5). We were particularly interested in the distribution of antibiotic-resistant genes (ARGs) in wild boars. Compared with Duroc pigs, the gut microbiome of wild boars had a significantly lower number of ARGs (wild boars vs. Duroc-JY, $P = 6.9 \times 10^{-4}$; wild boars vs. Duroc-SH, $P = 6.9 \times 10^{-4}$) (Supplementary Fig. 10a). The abundances of resistance classes that ARGs were classified into in wild boars and Duroc pigs were shown in Supplementary Fig. 10b. Duroc pigs had a high abundance of ARGs related to tetracycline, aminoglycoside, nucleoside, M-L-S (macrolide antibiotic, lincosamide antibiotic, and streptogramin antibiotic), and lincosamide. However, almost all classes of antibiotic resistance genes were less abundant in wild boars (Supplementary Fig. 10b and Supplementary Table 4).

## Discussion

In the current study, a reference gene catalog of the pig gut microbiome was generated covering more than 17.24 million full-length proteins clustered at 90% identity and 6339 microbial genomes were constructed. These datasets are comprehensive

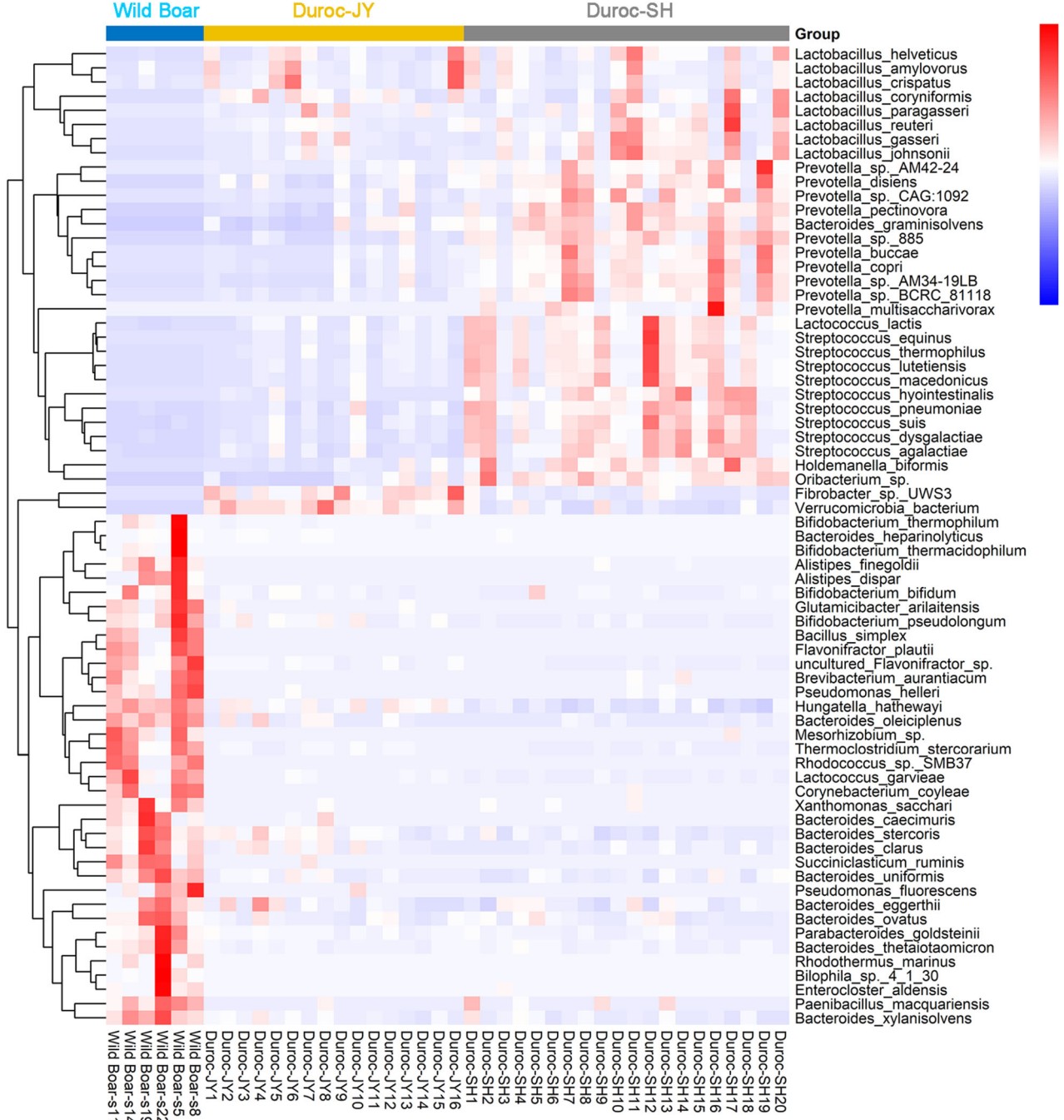

**Fig. 5 Bacterial species enriched in wild boars and commercial Duroc pigs, respectively.** Heatmap showed the parts of bacterial species enriched in wild boars and both two Duroc populations (Duroc-JY and Duroc-SH), respectively, at the significance threshold of Bonferroni-corrected *P*-value <0.01. All 180 significant bacteria species are listed in Supplementary Data 3.

resources for investigating the pig gut microbiome. The improved ratio of mapped sequence reads of the metagenomic sequencing datasets and several million new proteins based on UniProt TrEMBL database in the PIGC90 means that this gene catalog will greatly facilitate assembly-free metagenomic sequencing data analysis and the metatranscriptome and metaproteome analyses similar to that performed on host genomes and via transcriptomics analysis using mapping approaches[40,41].

Compared with the previously reported PGC catalog, the PIGC was constructed using a diversified landscape of samples not only from feces, but also from the lumen of different gut locations, and not only from domestic pigs worldwide, but also from wild boars.

This broad sample source significantly increased the diversity and representation of the PIGC (Fig. 2d). However, compared with the contribution of more than 170,000 specific genes from the domestic pig, the utilization of lumen samples from ileum and cecum, and the samples from wild boars did not contribute many specific species or functional categories to the metagenome (Supplementary Fig. 7). This may be due to a number of factors: (1) pan-genome of microbial species among sample sources; (2) limited annotation using the current database; (3) the effect of much lower sample numbers of ileum, cecum, and wild boars. The predicted gene (protein) number increased steadily following increasing sequencing depths. However, most of the genes in the

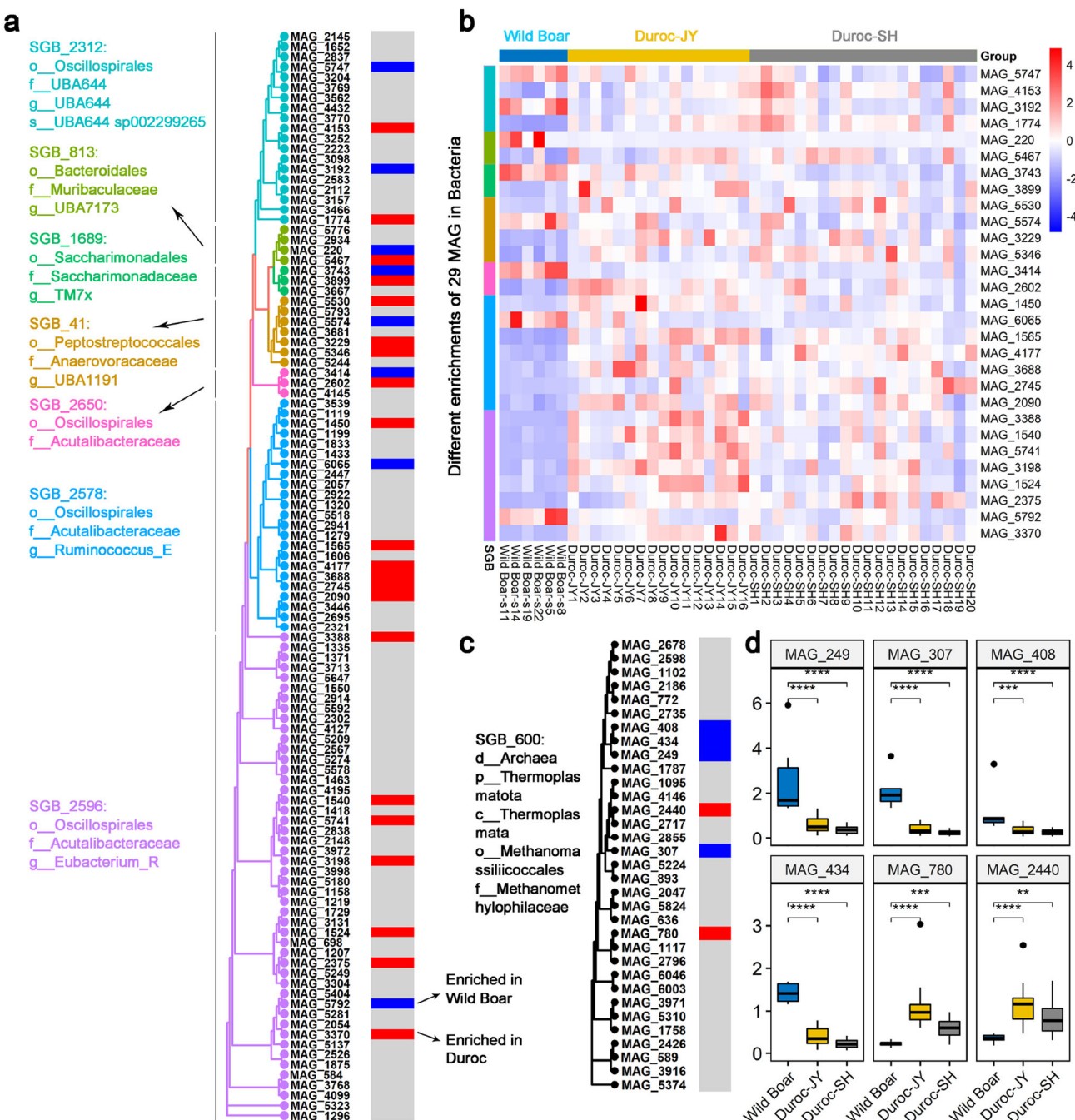

**Fig. 6 The species-level genome bins (SGBs) containing metagenome-assembled genomes (MAGs) showing different directions of enrichment in wild boars and Duroc pigs. a** The phylogenetic tree showing all MAGs from seven SGBs belonging to bacteria. The different colors distinguish each SGB. In each of these seven SGBs, some MAGs were enriched in wild boars (blue bars), some MAGs in Duroc pigs (red bars), and the others did not show significant difference between wild boars and Duroc pigs (gray). **b** Heatmap showing the different enrichments of the 29 MAGs from seven SGBs described above between wild boars and Duroc pigs. **c** The phylogenetic tree showing all MAGs from SGB_600, which belongs to Methanomethylophilaceae in Archaea. Four MAGs in this SGB were enriched in wild boars (blue bars) and two MAGs enriched in Duroc pigs (red bars). **d** Boxplots showing the different abundances of the six MAGs in the SGB_600 between wild boars ($n = 6$) and Duroc pigs (Duroc-JY: $n = 16$, Duroc-SH: $n = 20$). *$P < 0.05$, **$P < 0.01$, ***$P < 0.001$, ****$P < 0.0001$, two-tailed Wilcoxon test was used Boxplots show median, 25th and 75th percentile, the whiskers indicate the minima and maxima, and the points laying outside the whiskers of boxplots represent the outliers.

catalog had low abundance and prevalence within individual samples. This is similar to what was observed in the human gut and oral microbiome[42].

The 6339 MAGs are another highly valuable dataset resource generated by this study. Only a small portion of the MAGs represent previously sequenced microbial species (13.65%). Most of the MAGs appear to be novel species with sequenced relatives

at higher taxonomic levels. These MAGs significantly increase the number of available microbial reference genomes.

PIGC90 and constructed MAGs were used to compare the gut microbiomes of wild boars and highly commercialized Duroc pigs as an example of the implementation of PIGC90 and MAGs. Diversity of the gut microbiota is likely very important to animal health. Whether the high density in

capacity and formula feed in the modern pig industry affect the diversity of the pig gut microbiome is unknown. We did observe the higher α-diversity of the gut microbiome at the genus level in wild boars compared with that in Duroc pigs, especially in Duroc-SH. The gut microbiome of wild boars had a significantly higher abundance of *Bacteroides spp*. (Fig. 5a). *Bacteroides* play important roles not only in producing valuable nutrients and energy by breaking down food, but also in regulating immune abilities. For example, *B. uniformis* and *B. xylanisolvens* can utilize both dietary and endogenous glycans, along with the production of beneficial end products, such as short-chain fatty acids (SCFAs), for both the bacterium and the host[43,44]. *Bacteroides ovatus* is a representative of *Bacteroides* genus that has immune-regulatory abilities[45]. *Bifidobacterium spp*. were also significantly enriched in the gut microbiome of wild boars. *Bifidobacteria* can utilize a diverse range of plant-derived oligo- and polysaccharides that cannot be degraded by host[46] and produce SCFAs. SCFAs not only provide the energy for the survival of wild boars[47], but also play important roles in anti-inflammation and improving immunity[48]. The species from *Prevotella*, *Lactobacillus*, and *Streptococcus* were enriched in Duroc pigs. This is similar to the previous report based on 16S rRNA gene sequencing[49]. In particular, the relative abundance of *P. copri* in Duroc-JY and Duroc-SH pigs achieved 23.38% and 54.06%, respectively (Supplementary Fig. 9d). In a previous study, we showed the causative role of *P. copri* in host fat accumulation by pro-inflammation response, which relies on formula diets. Unlike wild boars, in the modern pig industry Duroc pigs are fed the commercial formula feed containing high levels of energy (3023 kcal/kg for Duroc-SH and 2960 kcal/kg for Duroc-JY) and protein to exploit growth potential, which induces an overgrowth of *P. copri* in the gut (Supplementary Fig. 9d). *S. suis*, a pathogen[50], was also enriched in Duroc pigs. Pigs raised on commercial farms with high density and fed commercial formula feed easily become sick. Consistent with the results of microbial composition, the functional capacities of the gut microbiome enriched in wild boars were mainly related to the metabolism of nutrients (amino acids, lipid, carbohydrate, and vitamin), which are essential functions to the life of both host and microbes. However, the functional capacities enriched in Duroc pigs were mainly related to genetic information processing, suggesting a high propagation of some bacterial species in the gut that may be induced by the high energy and protein diet. All together, these results suggest that the gut microbiome is greatly advantageous to the survival of wild boars in their natural free-living habitat.

In summary, the PIGC together with the constructed microbial genomes of the pig gut microbiome developed in the present study provide important and powerful resources for generating insights into the pig gut microbiome and for future metagenomic sequencing-based studies.

## Methods

**Animal management and sample collection**. A total of 470 pigs from a Mosaic population, a cross line, five domestic purebreds, and wild boars were gathered and used in this study (Supplementary Table 1). The mosaic population was constructed by random hybridization amongst four Western (Duroc, Landrace, Large White, and Pietrain) and four Chinese pig breeds (Bamaxiang, Erhualian, Laiwu, and Zang). All pigs were raised in a uniformed farm of Jiangxi Agricultural University in Nanchang, Jiangxi Province, and provided commercial formula feed twice daily. The feed contained 16% crude protein and 3100 kcal/kg digestible energy and 0.78% lysine as previously described[51]. Water was provided ad libitum from nipple drinkers. All piglets were weaned at the age of 42 days and the males were castrated at the age of 80 days. A total of 325 samples were collected from 301 $F_6$ pigs of this mosaic population, including 301 fecal samples at the age of 240 days, 10 fecal samples at the age of 25 days, and 10 cecal lumen and 4 ileal lumen samples upon slaughtering at the age of 240 days. Forty-nine sows from the Berkshire × Licha line and 14 purebred sows of Licha were included in this study.

Fecal samples were collected from all 63 sows, which were raised on a commercial farm in Dingnan, Jiangxi Province, and fed formula feed. Fecal samples from a total of 36 Duroc pigs from two commercial farms were collected and used in this study. Duroc pigs from Jiangyin Farm in Guangdong Province (Duroc-JY) were provided formula feed containing 2960 kcal/kg digestible energy and 15% crude protein, and Duroc pigs from Shahu Farm in Guangdong Province (Duroc-SH) were fed formula feed consisting of 3023 kcal/kg digestible energy and 17% crude protein. Fecal samples were collected from 27 Tibetan pigs from four farms, including 21 pigs from three high-altitude farms in Kangding, Sichuan Province, and 6 pigs from a farm in Nanchang, Jiangxi Province. Fecal samples from 29 Large White piglets at the age of 30 days were also included in this study. These piglets were raised in Liangyeshan farm, Fujian Province, and weaned at the age of 21 days. Eight adult wild boars were captured from the mountains in Jiangxi Province. Their feces and cecum luminal contents were harvested upon slaughter. Two samples were pooled for each of the jejunum, ileum, and cecum of the Laiwu pigs and used in metagenomic sequencing as previously described[37] and used to construct the gene catalog of the gut microbiome in this study. All fecal samples were collected from the rectum of experimental pigs. The luminal contents were separately harvested from the middle part of jejunum and ileum, and the bottom of the cecum within 30 min after slaughter. All experimental pigs were healthy and received no probiotic or antibiotic therapy within 2 months of sample collection. All animal work was conducted according to the guidelines for the care and use of experimental animals established by the Ministry of Agriculture of China. The project was also approved by Animal Care and Use Committee (ACUC) in Jiangxi Agricultural University.

**DNA extraction**. All samples were dipped in liquid nitrogen, and then transferred into a −80 °C freezer until use. DNA was extracted using the QIAamp Fast DNA Stool Mini Kit (Qiagen, Hilden, Germany) according to the manufacturer's instructions. The concentration and the quality of all DNA samples were measured using the NanoDrop-1000 and agarose gel electrophoresis.

**DNA library construction and metagenomic sequencing**. DNA libraries were constructed following the manufacturer's instructions (Illumina, San Diego, CA, USA), and index codes were added to attribute the sequences of each sample. The clustering of the index-coded samples was performed on a cBot Cluster Generation System according to the manufacturer's instructions. After cluster generation, the library preparations were sequenced on a Novaseq 6000 platform adopting a 150-bp paired-end sequencing strategy.

**Metagenome assembly**. Raw reads were filtered to remove adapter sequences and low-quality sequences using fastp (v0.19.4, --cut_by_quality3 -W 4 -M 20 -n 5 -c -l 50 -w 3)[52], and the reads mapped to the host genomic DNA by BWA MEM (v0.7.17-r1188)[53] were filtered out. Metagenome assembly was processed by MEGAHIT (v1.1.3)[54] using the options '--min-count 2 --k-min 27 --k-max 87 --k-step 10 --min-contig-len 500'. The sequence data of each sample were assembled individually. To make full use of the sequence reads and identify rare genes, the clean sequence data of all samples were aligned to the assembled contigs by Bowtie 2 (v2.3.4.1)[55] to acquire the unassembled reads. All unassembled reads of each sample in a pig population were then pooled and co-assembled with MEGAHIT (v1.1.3) using the same parameters for a single sample.

**Construction of the gene catalog**. A total of 201,051,463 assembled contigs were used for gene prediction by Prodigal (v2.6)[56] software. After removing the incomplete genes, those genes with a start and stop codon were retained for further analyses. To expand the gene catalog of the current study, the complete genes were integrated from 7,685,872 nonredundant genes in the gene catalog of the pig gut microbiome using 287 pig samples reported previously[29]. All complete genes were clustered at protein level following UniRef guidelines[36] at 100% (PIGC100), 90% (PIGC90), and 50% (PIGC50) of protein identity using CD-HIT (v4.8.1)[57]. Each member of the clusters in PIGC90 and PIGC50 was required to have at least 80% of sequence overlap with the longest (seed) sequence. All genes were aligned to the Uniprot TrEMBL database (https://www.uniprot.org/statistics/TrEMBL) to filter out those genes belonging to eukaryotes (except fungi).

**Taxonomic and functional annotation of genes**. The amino acid sequences of proteins in the catalog were aligned to the Uniprot TrEMBL by DIAMONG (v0.9.21.122)[58] with e-values ≤1e−5. The proteins that could not be aligned in the database were defined as unknown proteins. For those genes that were matched to distinguishable taxonomic groups (with multiple records of e-value ≤1e−5), the taxonomic classification was determined based on the lowest common ancestor algorithms by BASTA (v1.3)[59] at the thresholds of an alignment length >25, identity >80%, and shared by at least 60% of hits. Similarly, the genes that could not be classified to any taxa were defined as unknown taxa. The KEGG (Kyoto Encyclopedia of Genes and Genomes) annotation results were extracted with KOBAS (v3.0.3)[60] software (-t blastout:tab, -s ko). The eggNOG was annotated by aligning genes to eggNOG 5.0[61] database using eggnog-mapper (v2.0.1)[62], and carbohydrate-active enzymes (CAZymes) were annotated by aligning genes to dbCAN database (HMMdb V8)[63] with hmmscan program in HMMER (v3.1b2)[64].

Antibiotic resistance genes (ARGs) were annotated by alignment against the Comprehensive Antibiotic Resistance Database (CARD) using RGI (v5.1.1)[65]. The virulent protein sequences were identified by annotation with the Virulence Factor Database (VFDB)[66] using BLAST (v2.10.1+)[67]. For the aligned protein sequences, the annotated hit(s) with the highest score was used for the subsequent analysis[21,24,68].

**Genome reconstruction.** Genome reconstruction of gut microbes with metagenomic sequencing data was performed with the function modules of metaWRAP (v1.1.1)[69], which is a pipeline that includes numerous modules for analyzing metagenomic bins. Briefly, metagenomic bins (or metagenome-assembled genomes, MAGs) were constructed with contigs from single sample, and co-assembly contigs (co-assembled from unmapped reads of all samples in each pig population) in 500 metagenome data by two different binning algorithms of '--metabat2 --maxbin2'[70,71] in metaWRAP software. The default of the minimum length of contigs used for constructing bins with Maxbin2 and metaBAT2 were 1000 and 1500 bp, respectively. For the construction of metagenomic bins with co-assembled contigs by Maxbin2, the minimum length of contigs was set at 2500 bp due to memory limitation and time-consuming computation.

Refinement of MAGs was performed by the bin_refinement module of metaWRAP[69]. First, bin set AB was generated by combining bin set A produced by MaxBin2 and bin set B produced by MetaBAT2. CheckM (v1.0.12)[72] was used to estimate the completeness and contamination of the bins in sets A, B, and AB to choose the best one of each MAG with the highest scoring function S=Completion-5*Contamination value. To improve the quality of bins, metagenomic sequence reads were mapped to each bin, and then, reassembled with metaSPAdes[73] via the reassemble_bins module of metaWRAP. CheckM (v1.0.12) was re-run to estimate completeness and contamination of the final bins.

**Dereplication and species-level clustering of MAGs.** dRep (v2.2.3)[74] was used for dereplication of all 21,609 MAGs by two-step cluster. First, MAGs were divided into primary clusters using Mash at a 90% Mash ANI. Then, each primary cluster was used to form secondary clusters at the threshold of 99% ANI with at least 25% overlap between genomes. According to the criteria of quality evaluation by CheckM (v1.0.12), 6339 nonredundant MAGs were divided into medium-quality MAGs (more than 50% completeness and <5% contamination) and high-quality MAGs (more than 90% completeness and <5% contamination)[38,39]. The MIMAG standards set up by the Genomic Standards Consortium, which require the 5S, 16S, and 23S rRNA genes, and at least 18 tRNAs in MAGs[39] were also used to evaluate the quality of MAGs.

MAGs were clustered into species-level genome bins (SGBs) at the threshold of 95% ANI using the 'cluster' program in dRep (v2.2.3). SGBs containing at least one reference genome (or metagenome-assembled genome) in the Genome Taxonomy Database (GTDB, https://gtdb.ecogenomic.org/) were considered as known SGBs. And SGBs without reference genomes were considered as unknown SGBs (uSGBs)[31].

**Phylogenetic analysis and genome annotation of MAGs.** A total of 6339 representative MAGs were classified using the Genome Taxonomy Database Toolkit (GTDB-tk)[75]. All phylogenetic trees of the MAGs were built by PhyloPhlAn (v3.0.51)[76] and visualized using iTOL (v5.6.2)[77] or ggtree (v2.3.3.993)[78,79] in R package (v3.6.2). The genome annotation of MAGs, including the prediction of coding sequence (CDS), tRNA, and rRNA, was performed with Prokka[80] using the annotate_bins module of metaWRAP (v1.1.1).

**Estimation of the abundances of genes, taxa, function terms, and MAGs.** Clean reads of each sample were aligned to the gene catalog using BWA MEM (v0.7.17-r1188)[53]. The outputs were converted to BAM format by Samtools (v1.10)[81]. FeatureCounts (v2.0.1)[82] was then used to compute the number of successfully assigned reads. The abundances were normalized to fragments per kilobase of gene sequence per million reads mapped (FPKM)[83]. For each sample, the FPKM was calculated by the formula:

$$\text{FPKM} = \frac{numFragments}{\frac{geneLength}{1000} \times \frac{totalNumReads}{1,000,000}} \qquad (1)$$

where *numFragment* is the number of fragments mapped to a gene sequence; *geneLength* is the length of the gene sequence; and *totalNumReads* is the total number of mapped reads of a sample.

The abundances of microbial taxa, KEGG Orthology (KO), KEGG pathway, eggNOG Orthology, CAZyme, Antibiotic Resistance Ontology (ARO), and Virulence Factors (VF) were calculated by adding the abundances of all the members falling within each category. Salmon (v0.9.1) was used to quantify the abundance of each MAG in each sample with alignment-based mode[84].

**Statistical analysis**
*Rarefaction curve analysis.* Based on the gene profile table, random sampling was performed 100 times in all tested samples without replacement for a given number of sample size and the total number of nonredundant genes was calculated for each

sampling time. The sample size increased from 20 to 500 at the rate of 20 samples per step. The average of the 100 sampling times was plotted for each sample size.

*Contribution of sequencing depth to the number of nonredundant genes.* To explore the effect of sequencing depth on the capture of gut microbial genes, the correlation between the sequencing depth and the number of predicted genes was analyzed in 301 feces samples from $F_6$ pigs of the Mosaic population. The adjusted R-squared and P-values were calculated using a linear regression model in R package (v3.6.2). We further selected 20 fecal samples from the $F_6$ population with about 12.4-Gb sequencing data, and randomly drew 6.2-Gb reads from each sample for assembling, and predicting microbial genes. The number of predicted genes between two sequencing depths was compared with the Wilcoxon test by *ggpubr* in R package (v3.6.2).

*Computation of the mapping ratios of metagenomic sequencing reads.* For the PGC, 3,460,040 complete genes were extracted from the gene catalog and dereplicated at the protein level using CD-HIT (v4.8.1) as described above. A total of 76 metagenomic sequencing data obtained from diverse sample sources were downloaded from five public datasets. After quality control and removing the host genomic sequence using the methods described above, the clean reads of each sample were aligned to the PIGC90 and PGC90 by BWA MEM (v0.7.17-r1188). The number of mapped reads were counted using Samtools (v1.10). The ratio of mapped reads was calculated by computing the percentage of the mapped reads in the total number of reads obtained in each sample. The boxplots were visualized by *ggpubr* in R package (v3.6.2).

*Other statistical analyses.* The α-diversity of gut microbiota including the number of observed species and Shannon index, and principal coordinate analysis based on the Bray–Curtis distance were calculated by *vegan* in R package (v3.6.2). Comparison of the gut microbiome between wild boars and Duroc pigs was performed by two-tailed Wilcoxon test (pairwise comparison) or Kruskal–Wallis (multiple group comparison). Bonferroni-corrected $P < 0.01$ was set as the significance threshold for the comparisons of bacteria species, KEGG pathway and MAGs between wild boars and Duroc pigs. The results were visualized with the boxplots or heatmaps plotted by *ggpubr* and *pheatmap* in R package (v3.6.2), respectively. Text processing, information extraction and data statistics in pipeline for the construction of PIGC and MAGs were processed using R (v3.6.2), Python (v3.5.5), or Perl (v5.10.1) programs.

**Public data used.** The gene catalog of the pig gut microbiome from 287 samples (PGC) was downloaded from GigaScience Database with link http://gigadb.org/dataset/view/id/100187/token/F4CDHYruxobOKmsE. Five metagenomic sequencing datasets of the pig gut microbiome from the samples varying in countries, pig breeds, ages, and gut locations, were downloaded from NCBI with accession numbers ERP024389 ($n = 20$), SRP108960 ($n = 8$), SRP116179 ($n = 8$), and SRP188615 ($n = 40$), and used for evaluating the representation of microbial genes in the PIGC. All sources of public databases used in analysis pipeline can be available from GitHub Repository (https://github.com/zhouyunyan/PIGC).

**Reporting summary.** Further information on research design is available in the Nature Research Reporting Summary linked to this article.

## Data availability
Metagenomic sequencing data, microbial genes, and metagenome-assembled genomes from this study were submitted to China National GeneBank DataBase (CNGBdb) with accession code: CNP0000824. Source data are provided with this paper.

## Code availability
The codes for construction of the reference gene catalog and MAGs, and statistical analyses and visualization are available from the GitHub repository (https://github.com/zhouyunyan/PIGC)[85].

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

## Acknowledgements

This work was supported by National Natural Science Foundation of China (Nos. 31772579 and 31760654).

## Author contributions

L.H. and C.C. conceived and designed the experiments. Y.Z. performed most of the experiments. H.Y., J.W., H.F., X.X., J.G., H.J., and S.F. collected the samples and performed part of experiments. Y.Z. and C.C. analyzed the data. L.H., C.C., and Y.Z. wrote the paper.

## Competing interests

The authors declare no competing interests.
