## [Peer Review File · Nature Communications]

REVIEWER COMMENTS

Reviewer #1 (Remarks to the Author):

The paper by Chen et al has a lot of merit and deserves to be published in Nature Communications; however, there are many issues and flaws that need to be addressed first.

The paper is VAST and should be reduced in size. I suggest simply removing the sections "Comparison of the gut microbiomes of pig, human, dog, and mouse using gene catalogs", "Evaluating the effects of age, sex, breed, geographical distribution, and gut location on the gut microbiome", "Evolution of gut microbiome from wild boars to highly selected commercial Duroc pigs" - they are long, confusing and add very little to the paper. It is often not clear whether the results are based on the gene catalog or the MAGs, and often statements are not robustly backed up by evidence.

In my view this should be a simpler paper, about a new resource which is the pig gene catalog and the ~6000 MAGs.

Title: remove unexpected. Not sure why this result is unexpected?

29: nearly 600% - what does this mean? 600% of what?

41: close-to-complete - I am not sure this statement is justified

64: the authors miss several references: As well as Almeida and Pasolli, there is Nayfach (<https://www.nature.com/articles/s41586-019-1058-x>) - but in fact even these have been summarised in <https://www.biorxiv.org/content/10.1101/762682v1> so perhaps a single reference for the human data? And there is also cattle (<https://www.nature.com/articles/s41587-019-0202-3>) and chicken (<https://genomebiology.biomedcentral.com/articles/10.1186/s13059-020-1947-1>) which as agricultural species are arguably more relevant to this article

106: unprecedented - again I don't think this term is justified

In terms of the gene catalog I am not sure this has been constructed correctly. I believe it will contain many fragments and this will inflate the number of genes. It's also not clear whether redundancy is defined at the nucleotide level (which will inflate numbers again) or protein level.

As such, I recommend the following.

First, filter for genes that are estimated to be complete. If using Prodigal, then there should be a code in the FASTA header: partial=00 means complete. 10, 01, and 11 all predict truncated genes. Remove truncated genes.

Second, remove redundancy at protein level using dbCAN

Third, follow UniRef guidelines:

Using CD-HIT program, that can be done with following arguments

ID=100%: -c 1.0

ID=90%: -c 0.90 -n 5 -s 0.80

ID=50%: -c 0.50 -n 3 -s 0.80

So here we do successive dereplications of proteins - first at 100% identity; then at 90% then at 50%.

153: the PIGC napping rates vs PGC mapping rates are not that much higher (5-10%) suggesting the new data is not that important?

157-164: I could not understand this section

168: using NCBI NR for taxonomic classification is not a good idea as many sequences are misclassified. I recommend RefSeq or UniProt TREMBL

195: the authors cannot say that microbes interact with one another, this is simply a correlation analysis

254: please use standards for describing MAGs as set out in Bowers et al (<https://pubmed.ncbi.nlm.nih.gov/28787424-minimum-information-about-a-single-amplified-genome-misag-and-a-metagenome-assembled-genome-mimag-of-bacteria-and-archaea/>) - note this includes presence of 16S genes and tRNA genes

258: average size and N50 are meaningless as there is a range of completeness

271-272: I highly recommend GTDB-Tk to assign a taxonomy to the unknown MAGs

305: "pathogen" - this is a highly suspect use of the term. E coli is not always a pathogen. It is also not clear to me how the abundance of these, or the enrichment within the wild boar, are defined. I cannot see the evidence here that there are more pathogens in the boar, these statements need to be caveated!

427: accompanying paper - where is this paper?

520: merged and dereplicated - how?

536: I think the authors mean lowest common ancestor

539: which version of CAZy? Perhaps the authors used dbCAN? If so this should be referenced

581: the authors should use PhyloPhlAn 2 and also GTDB-Tk

658: I could not find the accession at the CNGBdb: <https://db.cngb.org/search/?q=CNP0000824>

Reviewer #2 (Remarks to the Author):

The authors report the construction of an integrated gene catalog of the pig gut microbiome that combines an already published catalog of 7.7M genes with sequencing data from 500 other samples: 452 fecal samples, 28 luminal samples from caecum (20 samples), ileum (6 samples), jejunum (2 samples), and 20 fecal samples from a previous study. The new gene catalog includes 48.3M genes, thus increasing the number of identified genes by 6.3 fold. The authors have applied a binning analysis method to assemble sequences and have reconstructed 6339 metagenome-assembled genomes. No doubt that this integrated catalog will be very useful for quantitative metagenomics and deep functional studies. The results are descriptive that is quite expected since the integrated catalog corresponds to the extension of an existing resource. Despite the importance of this new resource, the analysis and interpretation of the underlying biology is too limited. Many conclusions have already been reported: influence of age on microbiome composition; pairwise comparison of gut microbiome showing that pig is closer to humans than other species are; more shared items in a core microbiome considering the functions instead of the genes, reflecting redundancy of gene functions. The discussion section is more a repetition of results than a specific discussion aiming at highlighting important findings and perspectives. This report needs to be more focused on specific new knowledge provided by this integrated catalogue.

Main comments:

1. It is not appropriate to compare the integrated human gut microbiome gene catalogue published by Li et al. (2014) with the first pig catalogue. Indeed, it would be more appropriate to compare the first human catalogue published by Quin et al. (2010) who reported 3.3M genes from 124 European faecal samples to the first pig catalogue for which the authors identified 7.7M genes from 287 samples. The present report is thus more comparable to the first integrated human catalogue that included 9.9M genes from a combination of 249 newly sequenced samples of the Metagenomics of the Human Intestinal Tract (MetaHit) project with 1,018 previously sequenced samples. The authors could thus rely on these two types of catalogues to highlight the higher diversity of the pig gut microbiome compared to the human gut microbiome.
2. The new sequences provided many information on rare genes and associated assembled genomes. Have the authors produced a rarefaction curve to follow the adds of new genes according to the adds of new samples?
3. How relevant is it to study a core microbiome that takes into account faecal and luminal samples from different locations? It might be more biologically interesting to identify a core microbiome at each gut location.
4. What is the prevalence of antibiotic resistance genes? Are they present in domesticated pigs and wild boars as well?
5. The authors have included faecal samples and luminal samples, as well as faecal samples from domesticated pigs and wild boars. We strongly miss the specific contribution of these different data

sets to the whole catalogue. We would expect an extended and consolidated gene catalogue of the faecal microbiome of domesticated pigs together with the additional gene catalogue provided by wild boars' microbiome.

6. Since the authors have collected samples at different locations from same individuals, it would have been very interesting to discuss the relevance of considering the faecal microbiome as a proxy of the whole gut microbiome.

7. Microbiome diversity is likely very important to preserve for animal health and resilience. How much diversity have the domesticated pigs already lost (or not lost) compared to wild boars? This important issue should be better presented and discussed.

8. The animal sampling is associated with many parameters. The parameter distribution (age, farm, etc.) across the sampling population looks very unbalanced except for the sex ratio. More than half of samples come from the same farm with animals being between 210 and 240 days of age. The risk of confounding effects for the data analysis should be better presented and discussed. The animal metadata have to be precised: feed diet, antibiotics supplementation, age at weaning for the sampling at 25 and 30 days of age.

9. The study on the influence of the age on microbiome composition provides obvious results because the authors compare 25-day-old pigs to 240-day-old pigs. The young piglets were either not yet weaned or just weaned, meaning that the first round of microbial diversification has not already ended. This analysis does not provide any new information on the dynamic maturation of the gut microbiome over life.

10. For the comparison between domesticated pigs and wild boars, why have the authors compared Duroc pigs and wild boars? The experimental design is not appropriate for an evolution study as mentioned in the text. This is more a comparative study. The findings are very interesting and should be related to what is referred to as the pathobiome.

11. The authors state that the pig is an ideal model. This is an exaggerated statement. Even if pig is considered as a very good model for humans, that is not an ideal model.

October 8, 2020

We carefully checked the comments and revised the paper by point to point. The revisions on phrase and words were shown in the manuscript, and all revisions were highlighted in blue. The point-by-point responses to the concerns are listed as follows.

Reviewer #1 Comments:

1. The paper is VAST and should be reduced in size. I suggest simply removing the sections "Comparison of the gut microbiomes of pig, human, dog, and mouse using gene catalogs", "Evaluating the effects of age, sex, breed, geographical distribution, and gut location on the gut microbiome", "Evolution of gut microbiome from wild boars to highly selected commercial Duroc pigs" - they are long, confusing and add very little to the paper. It is often not clear whether the results are based on the gene catalog or the MAGs, and often statements are not robustly backed up by evidence.

In my view this should be a simpler paper, about a new resource which is the pig gene catalog and the ~6000 MAGs.

Response: We appreciate the reviewer's valuable suggestions, and we revised the manuscript accordingly. We deleted the sections of "Comparison of the gut microbiomes of pig, human, dog, and mouse using gene catalogs", "Evaluating the effects of age, sex, breed, geographical distribution, and gut location on the gut microbiome" from the manuscript. As an example of the implementation of the PIGC and MAGs, we still keep the section about "Comparison of gut microbiome between

wild boars and highly selected commercial Duroc pigs”, but we revised the description of this section and focused on: 1) the comparison of the gut microbiome between wild boars under natural free-living and Duroc pigs living in standard commercial farms (high density in captivity and fed formula feeds that contain high levels of protein and energy) at the both species and strain levels using the PIGC and MAGs; 2) Comparison of functional capacities of the gut microbiomes between wild boars and Duroc pigs. These made the manuscript more focusing. We hope the present version may satisfy the reviewer’s concern.

2. Title: remove unexpected. Not sure why this result is unexpected?

Response: Thank for the suggestion, and we have deleted “unexpected” from the title. To conform the requirement of the journal, we have also abbreviated the title as “Enhanced catalogs of microbial genes and metagenome-assembled genomes from the pig gut microbiome”.

3. 29: nearly 600% - what does this mean? 600% of what?

Response: we intend to show that the gene number in the gene catalog constructed in this study has increased near 6 times than that number reported before. We re-formulated the abstract and deleted this way of expression from the manuscript.

4. 41: close-to-complete - I am not sure this statement is justified

Response: We deleted this word from manuscript.

5. 64: the authors miss several references: As well as Almeida and Pasolli, there is Nayfach (<https://www.nature.com/articles/s41586-019-1058-x>) - but in fact even these have been summarized in <https://www.biorxiv.org/content/10.1101/762682v1> so perhaps a single reference for the human data? And there are also cattle (<https://www.nature.com/articles/s41587-019-0202-3>) and chicken (<https://genomebiology.biomedcentral.com/articles/10.1186/s13059-020-1947-1>) which as agricultural species are arguably more relevant to this article

Response: The reviewer's comments and suggestions are realistic and helpful. We have added these references to the manuscript.

6. 106: unprecedented - again I don't think this term is justified

Response: we deleted "unprecedented" from this sentence.

7. In terms of the gene catalog I am not sure this has been constructed correctly. I believe it will contain many fragments and this will inflate the number of genes. It's also not clear whether redundancy is defined at the nucleotide level (which will inflate numbers again) or protein level.

As such, I recommend the following.

First, filter for genes that are estimated to be complete. If using Prodigal, then there should be a code in the FASTA header: partial=00 means complete. 10, 01, and 11 all predict truncated genes. Remove truncated genes.

Second, remove redundancy at protein level using dbCAN

Third, follow UniRef guidelines:

Using CD-HIT program, that can be done with following arguments

ID=100%: -c 1.0

ID=90%: -c 0.90 -n 5 -s 0.80

ID=50%: -c 0.50 -n 3 -s 0.80

So here we do successive dereplications of proteins - first at 100% identity; then at 90% then at 50%.

Response: We appreciate the reviewer for these very detailed and useful comments, as well as the recommendation of tools and programs for improving the construction of gene catalog. We have re-constructed the gene catalog accordingly and updated the results. Please see the detailed information in the revised manuscript.

The brief introduction about analysis is described in the following:

First, Prodigal (v2.6) was used to predict genes of all assembled contigs from 500 samples, and extract the complete genes with a start and stop codon (partial=00).

Second, to expand the gene catalog of our current study, we integrated 3,460,040 complete genes in the gene catalog of the pig gut microbiome previously reported using 287 pig samples.

Third, a total of 126,545,050 complete gene were clustered at protein level using CD-HIT (v4.8.1) following UniRef guidelines:

- at 100% sequence identity;

- at 90% sequence identity;

and – at 50% sequence identity.

This generated three gene catalogs containing 48,697,887 (PIGC100), 17,237,052

(PIGC90), and 7,246,447 (PIGC50) non-redundant genes, respectively.

We removed redundancy by CD-HIT, which also do dereplication at the protein level.

Several other studies also used CD-HIT to do dereplication after gene prediction in the construction of gene catalog. For example:

Li, J., et al. An integrated catalog of reference genes in the human gut microbiome. *Nat Biotechnol* 32, 834-841 (2014).

Xiao, L., et al. A catalog of the mouse gut metagenome. *Nat Biotechnol* 33, 1103-1108 (2015).

Xiao, L., et al. A reference gene catalogue of the pig gut microbiome. *Nat Microbiol* 1, 16161 (2016).

8. 153: the PIGC mapping rates vs PGC mapping rates are not that much higher (5-10%) suggesting the new data is not that important?

Response: Here we present the good representation of the PIGC catalog. Better ratios of mapped sequence reads were obtained with PIGC90 (ranging from 87.03% to 97.83%) compared with PGC90 (ranging from 54.65% to 88.72%) (**Supplementary Fig. 4**)

9. 157-164: I could not understand this section

Response: We intended to show the completeness of the PIGC catalog and further explain why the mapping rate of sequence reads can achieve more than 95% using this catalog. As it may bring confusing, we deleted these sentences from manuscript.

10. 168: using NCBI NR for taxonomic classification is not a good idea as many sequences are mis-classified. I recommend RefSeq or UniProt TREMBL

Response: According to the reviewer's recommendation, we used UniProt TrEMBL for taxonomic classification.

11. 195: the authors cannot say that microbes interact with one another, this is simply a correlation analysis.

Response: We agree with this comment. It does only mean the correlation between bacterial species in their abundances. However, there may exhibit some real clues of

interaction among them, but we need more works to find out. We have deleted this analysis from the manuscript.

12. 254: please use standards for describing MAGs as set out in Bowers et al (<https://pubmed.ncbi.nlm.nih.gov/28787424-minimum-information-about-a-single-amplified-genome-misag-and-a-metagenome-assembled-genome-mimag-of-bacteria-and-archaea/>) - note this includes presence of 16S genes and tRNA genes

Response : Following this recommendation, we used the standards setting up by Bowers et al. (2017) for describing MAGs.

A final set of 6,339 non-redundant MAGs were obtained (**Supplementary Table 4**). Among them, 4,981 MAGs satisfied the medium-quality criteria (more than 50% completeness and less than 5% contamination), 1,358 MAGs showed high-quality (more than 90% completeness and less than 5% contamination)(**Supplementary Fig. 8a**)^{35,36}. Most of the high-quality MAGs had contig number < 250 and ≥ 18 of the standard tRNAs (**Supplementary Fig. 8b, c**). Of the 1,358 high-quality MAGs, 36 MAGs had the 5S, 16S and 23S rRNA genes together with at least 18 tRNAs, and conformed to the MIMAG standards for the ‘high quality’ MAG set by the Genomic Standards Consortium³⁶.

13. 258: average size and N50 are meaningless as there is a range of completeness

Response: Ok, we agree and we have deleted the description of the average size and N50 of MAGs from the results in the revised manuscript.

14. 271-272: I highly recommend GTDB-Tk to assign a taxonomy to the unknown MAGs

Response: According to the reviewer's recommendation, we used GTDB-Tk to perform taxonomic assignment.

15. 305: "pathogen" - this is a highly suspect use of the term. E coli is not always a pathogen. It is also not clear to me how the abundance of these, or the enrichment within the wild boar, are defined. I cannot see the evidence here that there are more pathogens in the boar, these statements need to be caveated!

Response: We thank the reviewer for indicating that E. coli is not always a pathogen. We agree on this comment, and to avoid unnecessary misunderstanding, we re-did the analysis using the taxonomic annotation data of the PIGC90 based on UniProt TREMBL, and re-wrote this result as follows:

We focused on those bacterial species enriched in wild boars or Duroc pigs using the PIGC90. We discovered that the gut microbiome of wild boars had significantly higher abundance of the bacterial species from *Bacteroides* (10 species) and *bifidobacterium* (four species), *Hungatella hathewayi*, and *Alistipes* (two species) (Fig. 5a, Supplementary Table 6). However, the species from *Prevotella* (11 species) and *Lactobacillus* (8 species) which are associated with porcine fat accumulation and lean meat percentage (reported in another paper under review) were enriched in both Duroc populations (e.g. the relative abundance of *P. copri* achieved 23.38% and 54.06%

in Duroc-JY and Duroc-SH, respectively) (Fig. 5a and Supplementary Fig. 9c).

16. 427: accompanying paper - where is this paper?

Response: we submitted another manuscript about *P. copri* increasing host fat accumulation. This manuscript is also under review, so we noticed that this content is under review. Please see line 358, page 17.

17. 520: merged and dereplicated - how?

Response: we used the method that the reviewer recommended to merge the data and dereplication. The more details were described as the following:

After removing the incomplete genes, only those genes with a start and stop codon were retained for further analyses. To expand the gene catalog of the current study, we integrated those complete genes from 7,685,872 non-redundant genes in the gene catalog of the pig gut microbiome previously reported using 287 pig samples³². All complete genes were clustered at protein level following UniRef guidelines³³ at 100% (PIGC100), 90% (PIGC90) and 50% (PIGC50) amino acid identity using CD-HIT (v4.8.1)⁵². Each member of PIGC90 and PIGC50 clusters was required to have at least 80% of sequence overlap with the longest (seed) sequence.

18. 536: I think the authors mean lowest common ancestor

Response: Yes, exactly.

We think both “last common ancestor” and “lowest common ancestor” are right.

According to this comment, we modified “last common ancestor” to “lowest common

ancestor”. Last Common Ancestor taxonomy estimation is also known as lowest common ancestor (Wood & Salzberg, 2014) or consensus (Caporaso et al., 2010) taxonomy estimation. In the reference for BASTA software (Kahlke, T., Ralph, P.J. & Price, S. BASTA–Taxonomic classification of sequences and sequence bins using last common ancestor estimations. *Methods in Ecology and Evolution* 10, 100-103 (2018).), it is described as Last Common Ancestor (LCA).

19. 539: which version of CAZy? Perhaps the authors used dbCAN? If so this should be referenced

Response: Thank you for indicating this. Carbohydrate-active enzyme (CAZyme) were annotated by aligning to dbCAN database (HMMdb-V8) by hmmscan program in HMMER (v3.1b2). and we added the references.

20. 581: the authors should use PhyloPhlAn 2 and also GTDB-Tk

Response: We have followed this recommendation and used GTDB-Tk to perform taxonomic classification. PhyloPhlAn was updated to PhyloPhlAn (v3.0) (Asnicar et al., Precise phylogenetic analysis of microbial isolates and genomes from metagenomes using PhyloPhlAn 3.0. *Nat Commun.*, 2020, 11: 2500), so we use PhyloPhlAn (v3.0) to perform phylogenetic analysis.

21. 658: I could not find the accession at the CNGBdb:
<https://db.cngb.org/search/?q=CNP0000824>

Response: We checked the CNGBdb. It seems now that we can find the accession at

this website. Maybe the reviewer can try again. If there is any problem, we can contact the CNGBdb

Reviewer #2 (Remarks to the Author):

No doubt that this integrated catalog will be very useful for quantitative metagenomics and deep functional studies. The results are descriptive that is quite expected since the integrated catalog corresponds to the extension of an existing resource. Despite the importance of this new resource, the analysis and interpretation of the underlying biology is too limited. Many conclusions have already been reported:

influence of age on microbiome composition; pairwise comparison of gut microbiome showing that pig is closer to humans than other species are; more shared items in a core microbiome considering the functions instead of the genes, reflecting redundancy of gene functions. The discussion section is more a repetition of results than a specific discussion aiming at highlighting important findings and perspectives. This report needs to be more focused on specific new knowledge provided by this integrated catalogue.

Response: We appreciate the reviewer for these valuable comments and suggestions. According to these comments, we revised the manuscript essentially. We deleted the contents about influence of the age on microbiome composition, pairwise comparison of gut microbiome showing that pig is closer to humans than other species are, and so

on, and focused more on specific new knowledge as listing in the following:

1) We focused more on the pig integrated gene catalog (PIGC) of gut microbiome constructed in this study, which was comprised of more than 17 million complete genes. Because the significantly higher depth of metagenomic sequencing and more extensive sample sources in this study than that in previous reports, we investigated the contribution of sequencing depth and extensive sample sources to gene content of the PIGC.

2) More focusing on the reconstruction of microbial genomes from gut metagenomic sequencing data. More than 6,300 non-redundant MAGs were obtained. These MAGs were clustered into 2,309 SGBs, 86.38% of which are unknown SGBs. This has provided a new dataset for the analyses of pig gut microbiome at the strain level.

3) As the example about the implementation of the PICG and MAGs, we compared the gut microbiome between wild boars and commercial Duroc pigs, which represented the pigs raised in two extremely different conditions: natural free-living and standard commercial farms (high density in captivity and fed formula feeds that contain high levels of protein and energy). The results provide important clues about the changes of pig gut microbiome from wild boars to highly commercial pigs.

1. It is not appropriate to compare the integrated human gut microbiome gene catalogue published by Li et al. (2014) with the first pig catalogue. Indeed, it would be more appropriate to compare the first human catalogue published by Quin et al. (2010) who reported 3.3M genes from 124 European faecal samples to the first pig

catalogue for which the authors identified 7.7M genes from 287 samples. The present report is thus more comparable to the first integrated human catalogue that included 9.9M genes from a combination of 249 newly sequenced samples of the Metagenomics of the Human Intestinal Tract (MetaHit) project with 1,018 previously sequenced samples. The authors could thus rely on these two types of catalogues to highlight the higher diversity of the pig gut microbiome compared to the human gut microbiome.

Response: We agreed with this comment. However, we can observe the higher diversity of the pig gut microbiome compared to the human gut microbiome even under the comparison between the first pig catalogue and the integrated human catalogue. We think the result should be more remarkable if the present catalog was compared to the integrated human catalogue.

According to two reviewers' suggestion, this comparison between gene catalogs of the human gut microbiome and the pig gut microbiome were deleted from the manuscript.

2. The new sequences provided many information on rare genes and associated assembled genomes. Have the authors produced a rarefaction curve to follow the adds of new genes according to the adds of new samples?

Response: Thanks for the reviewer's valuable suggestions. We have made a rarefaction curve which we had not included in the manuscript. The reason is that the saturation point of gene numbers identified following the sample size really depended on the completeness of gene catalog. The sample types of the present study covered

various breeds, ages, geographical places, gut locations and wild/domestic animals, and an average sequencing depth achieved 11.46 Gb/sample (21 Gb for the highest one), so the PIGC is more completeness than that reported previously. When we counted the new gene numbers by rarefying the sample size, the sequence reads of each sample were mapped to the PIGC to count the gene numbers identified in this sample. The saturation point at 100 samples just means that the increased genes had low prevalence and abundance when the sample size was > 100 . If different sample sizes were used to construct the gene catalog, we think the gene number should not achieve saturation. For example, the numbers of complete gene for PGC (287 samples) and PIGC (787 samples) were 3,460,040 and 17,237,052, respectively.

Based on the reviewer's suggestion, we include this rarefaction curve in the revised manuscript. In addition, we systematically evaluated the contribution of sequencing depth to the gene number identified.

3. How relevant is it to study a core microbiome that takes into account faecal and

luminal samples from different locations? It might be more biologically interesting to identify a core microbiome at each gut location.

Response: We defined the core bacterial species and functional capacities of the gut microbiome at the threshold of existing in > 90% samples. We expected to see those bacteria having the broad distribution cross ages, gut locations and breeds (populations) of pigs tested in this study. The core microbiome cross ages, gut locations and breeds may provide clues that these microbes have been adapted with the host for a long time, and some of them might be essentially important for the host. We also agreed with the reviewer's suggestion about a core microbiome at each gut location, which should be related to biological functions of each location. And according to this comment, we analyzed the core microbiome at each gut location. Please see Fig. 3b and Supplementary Fig. 7.

Ten bacterial species including *Lactobacillus reuteri*, *Lactobacillus johnsonii*, *Lactobacillus amylovorus*, *Ruminococcaceae bacterium*, *Escherichia coli*, *Prevotella copri*, *Bacteroides fragilis* and *Streptococcus suis* are in the top 20 lists of all feces, ileum and cecum samples. However, there were 16 bacterial species only in the top 20 list of one gut location. This result suggested the significant difference of the relative abundances of the same bacterial species in different gut locations. Different relative abundances in different gut locations should be associated with biological functions of each gut location.

Fig. 3b, The top 20 bacterial species in relative abundances in ileum lumen, cecum lumen and feces, respectively. The yellow color indicates the top 20 species in all three gut locations, and the colors corresponding to boxplots show the top 20 species specific to each gut location.

Supplementary Fig. 7. Comparison of core functional capacities of the gut microbiome at different gut locations and between wild boars and domestic pigs.

All these results above were also shown in the revised manuscript. Please see page 8 to 10.

4. What is the prevalence of antibiotic resistance genes? Are they present in domesticated pigs and wild boars as well?

Response: The details about the distribution of antibiotic resistance genes in domestic

pigs and its association with gut microbiota are described in a more topic paper which we have just submitted. In this manuscript, we want to provide a general description of the distribution of antibiotic resistance genes in wild boars and highly selected commercial Duroc pigs, showing how artificial commercial feeding affects the distribution of antibiotic resistance gene in the swine gut microbiome. The results are shown in the following and the revised manuscript (page 14-15).

Compared with that of Duroc pigs, the gut microbiome of wild boars had significantly lower number of ARGs ($P < 0.005$) (**Supplementary Fig. 10a**). The abundance of resistance classes of ARGs in wild boars and Duroc pigs is shown in **Supplementary Fig. 10b**. Duroc pigs had high abundances of ARGs resisting antibiotics of tetracycline, aminoglycoside, nucleoside, M-L-S (macrolide antibiotic, lincosamide antibiotic and streptogramin antibiotic) and lincosamide. However, almost all classes of antibiotic resistance in wild boars had lower abundance (**Supplementary Fig. 10b and Supplementary Table 8**).

5. The authors have included faecal samples and luminal samples, as well as faecal samples from domesticated pigs and wild boars. We strongly miss the specific contribution of these different data sets to the whole catalogue. We would expect an extended and consolidated gene catalogue of the faecal microbiome of domesticated pigs together with the additional gene catalogue provided by wild boars' microbiome.

Response: We agree with the reviewer's suggestion, and analyzed the specific contribution of these different data sets to the whole catalogue. The results are shown in the following and the revised manuscript (Page 7- 8).

Among the 17,237,052 protein clusters (non-redundant genes), 2,843,245 genes are sample source-specific (16.5%). Feces-specific genes from adult domestic pigs occupy the most part of the sample source-specific genes (94.0%). This should be due to the largest sample size (n = 427). The samples from piglets contributed 78,565 specific genes. Except feces samples, the samples from different gut locations (small intestine and cecum lumen) only contributed 168,737 non-redundant genes (including small intestine lumen-specific, cecum lumen-specific, and small intestine and cecum lumen-shared genes in domestic pigs and wild boars) (**Fig. 2e**). To our knowledge, this was the first study including the cecum lumen and feces samples from wild boars for constructing the gene catalog of pig gut microbiome. These samples provided 95,302 wild boar-specific non-redundant genes (**Fig. 2e**).

6. Since the authors have collected samples at different locations from same individuals, it would have been very interesting to discuss the relevance of considering the faecal microbiome as a proxy of the whole gut microbiome.

Response: Thanks for the comments and we revised these accordingly in both result and discussion sections.

The samples from different gut locations (small intestine and cecum lumen) contributed 169,125 non-redundant genes (result section, page 8). At the taxonomy level, although six bacterial species are in the top 20 lists of all feces, ileum and cecum samples, they were ranked in different orders in different sample types. This result suggested the significant difference of the relative abundances of the same bacterial species in different gut locations. Different relative abundances in different gut locations should be associated with biological functions of each gut location. We detected few gut location-specific species, but many species showed different

abundances in different gut locations (Supplementary Fig. 7), e.g. *Clostridium* (*Clostridium butyricum* and *Clostridium perfringens*) had high abundance in ileum lumen samples (result section, page 9). We also discussed this result in the discussion section (page 15).

7. Microbiome diversity is likely very important to preserve for animal health and resilience. How much diversity have the domesticated pigs already lost (or not lost) compared to wild boars? This important issue should be better presented and discussed.

Response: This is really a very important issue. Based on our data, we did not observe the significant difference in the α -diversity of the gut microbiome between wild boars and Duroc pigs at the species level although compared to Duroc-JY, significantly lower Shannon index were observed in Duroc-SH pigs (but no significant difference between wild boars and Duro-SH), which should be resulted by extremely high abundance of *P. copri* (the overgrowth of *P. copri* in the gut) (Supplementary Fig. 9b and c). We showed this result more in the result section and discussed it in the discussion section. Please see page 13 and 17.

8. The animal sampling is associated with many parameters. The parameter distribution (age, farm, etc.) across the sampling population looks very unbalanced except for the sex ratio. More than half of samples come from the same farm with animals being between 210 and 240 days of age. The risk of confounding effects for

the data analysis should be better presented and discussed. The animal metadata have to be precised: feed diet, antibiotics supplementation, age at weaning for the sampling at 25 and 30 days of age.

Response: We appreciate the reviewer for the concerns about animal sampling. Our sampling plan was to cover various breeds (wild boars, domesticated aboriginal breeds, highly selected commercial breeds and their crosses), ages (postnatal day 25 and 240), geographical places (Tibetan, Guangdong, and so on), and gut locations (ileum, cecum and feces). But in the actual experiment, we obtained a lot more samples from our designed mosaic population from 8 breed cross. In our construction of the PIGC catalog, it contained two parts of the samples. The first part of the samples was collected in this study from the pigs from more than eight farms although as the reviewer indicated, more than half of samples came from the same farm (301 pigs). Because we also combined the gene catalog reported previously (Xiao et al., 2016), the second part of the samples was from 287 pigs of various ages and breeds, on different diets in French, Danish and Chinese farms, which were used in the previous report (Xiao et al., 2016). In this way, we think we have presented as most diversified samples as we could for the microbial genes and MAG catalog in pigs.

To avoid the risk of confounding effects on the comparison of the gut microbiome among pig populations, the comparison was only performed between wild boars and Duroc pigs, where limited sampling parameters were involved.

Here we agree with the reviewer that the effect of different sample size for each type of samples (e.g. cecum lumen samples vs. feces) on the number of the sample

type-specific genes can't be excluded. We indicated this in the revised manuscript.

Please see Line 152-154, page 8.

For the samples collected in this study, we have detailed descriptions in the section of “Animal management and sample collection” for feed diet, antibiotics supplementation, and age at weaning

9. The study on the influence of the age on microbiome composition provides obvious results because the authors compare 25-day-old pigs to 240-day-old pigs. The young piglets were either not yet weaned or just weaned, meaning that the first round of microbial diversification has not already ended. This analysis does not provide any new information on the dynamic maturation of the gut microbiome over life.

Response: We agree with this comment. The comparison did only provide very limited information and the results about the influence of the age on microbiome composition were removed from the revised manuscript.

10. For the comparison between domesticated pigs and wild boars, why have the authors compared Duroc pigs and wild boars? The experimental design is not appropriate for an evolution study as mentioned in the text. This is more a comparative study. The findings are very interesting and should be related to what is referred to as the pathobiome.

Response: Yes, we agree with the reviewer. The comparison between wild boars and Duroc pigs is not enough for evolution study. The wild boars and Duroc pigs were separately represented the pigs raised in two extremely different conditions: for wild

boars, free-living and harvesting scavenge grass, roots and fruits as the foods for survival; and for Duroc pigs, a standard farm condition of modern pig industry with high density in capacity and providing high energy and protein formula diet. The comparison between these two types of pigs on their gut microbiome provide important clues about the changes of pig gut microbiome from wild boars to highly commercial pigs. Following the reviewer's suggestions, we have revised the manuscript (Page 12-15).

11. The authors state that the pig is an ideal model. This is an exaggerated statement. Even if pig is considered as a very good model for humans, that is not an ideal model.

Response: We agree with this comment and delete "ideal" from the manuscript.

All authors appreciate the editor and reviewers again for these invaluable comments and suggestion. We look forward to hearing from you at your earliest convenience. Your kind consideration for publication in *Nature Communications* will be greatly appreciated.

Please inform me if further information is required.

Yours sincerely

Prof. Lusheng Huang

President and Professor, Jiangxi Agricultural University
Member, Chinese Academy of Science (CAS)
Director, National Key Laboratory for Swine Genetic Improvement and Production Technology,
President, National Committee for Farm Animal Genetic Resources, China
NanChang, 330045, P.R.China.
Tel: 0086 791 3813080,
Fax: 0086 791 3900189

REVIEWER COMMENTS

Reviewer #1 (Remarks to the Author):

I am happy that the authors have responded to my comments

Reviewer #2 (Remarks to the Author):

The authors have thoroughly revised the article according to the comments. However, the manuscript is still too long and not sharp enough.

Main comments are listed below.

1) The introduction is actually confuse. It does not highlight well that the extended catalog provides an added value on the one hand in the number of non-redundant genes and on the other hand on the reconstruction of potential full bacterial genomes. Both are important and the number of reconstructed genomes should not shadow the importance of the increased gene number. The authors should also recall somewhere that all potential in silico reconstructed genomes need a biological confirmation of the existence of the corresponding microbial entity.

Line 40: what do the authors mean by “complex behaviors”?

Line 42: the authors wrote “Most of these studies focused on microbes with sequenced genomes”. Indeed, is it true? Most studies rely on available annotations that are often connected to partial genome sequences. Could the authors be more specific in this statement?

Line 49: suggestion to replace “host phenotypes (diseases)” by “host phenotypes and diseases”. All associated phenotypes are not related to diseases.

Lines 52 -54: “Reference genomes of microbiota are essential resources for understanding the functional role of specific microbes in the microbiome and for quantifying their abundance in metagenomes”. Quite challenging to put in the same sentence microbiota, microbiome and metagenomes. The authors might refer to Berg et al., 2020 for instance who have proposed to clarify the definition of each term (<https://pubmed.ncbi.nlm.nih.gov/32605663/>).

Again, since the first gene catalog for pigs was produced from only 287 fecal samples, it is not very appropriate to compare it with the integrated human catalog. The authors could instead promote

the idea that their aim was to build an integrated catalog for the pig as it had been done in humans. The integrated catalog in humans added rare genes mostly.

Lines 75-88: this ending paragraph is more a long summary of the results than a presentation of the main objectives of the study. In addition, the relevance to combine data from many fecal DNA (472) with cecum lumen (20), ileal lumen (6) and jejunum lumen (2) is not very clear. If the aim is to use a unique catalog to study any gut locations, that has to be stated in the aims of the study.

2) Results

Lines 134-144: was it not expected that the higher the sequencing depth the higher the gene number? Is that an original result that needs to be presented with so many details?

That is interesting to report the limited number of added genes provided by young pigs (before weaning, limited microbial diversification), gut sites and wild boars. It would be interesting to know whether these specific genes are rare or in high abundance in the samples, they came from.

If the reviewer understands well, the authors reported that the Duroc-JY pigs seem closer to the wild boars than to the Duroc-SH (cf. supp. Fig. 8). Are there hypotheses for this finding? Was it expected that the alpha diversity of the fecal DNA from wild boars was not higher than from the highly selected pigs?

Supp. Table 6: add the unit for the mean values (columns F, G, H)

Line 270: remove "reported in another paper under review"

Figure 1: explain all abbreviations and detail that the % identity is at the amino acid level

Line 817; replace piglet by piglet

Figure 3: Prevotella species were not found in the predominant core bacteria taxa in fecal DNA whereas they were identified as predominant in Duroc pigs for instance (e.g. Prevotella copri). Could this finding be discussed?

Title of supplementary table 8: replace "Comparison of antibiotic resistance between wild boars and Duroc pig" by Comparison of antibiotic resistance gene prevalence between wild boars and Duroc pigs"

Discussion

It is often reported that intensive selection has led to a decrease in domestic animal biodiversity and that wild boars are more prone to efficiently face pathogens. Have the data provided any clues relating to these general statements? Comparison between Duroc pigs and wild boars could contribute to discuss this issue.

Reviewer #1 (Remarks to the Author):

I am happy that the authors have responded to my comments.

Reviewer #2 (Remarks to the Author):

Comment 1: 1) The introduction is actually confused. It does not highlight well that the extended catalog provides an added value on the one hand in the number of non-redundant genes and on the other hand on the reconstruction of potential full bacterial genomes. Both are important and the number of reconstructed genomes should not shadow the importance of the increased gene number. The authors should also recall somewhere that all potential in silico reconstructed genomes need a biological confirmation of the existence of the corresponding microbial entity.

Response: We agree with the reviewer's helpful comments and suggestions, and have accordingly revised the introduction section of the manuscript, highlighted the values of both non-redundant gene catalog and reconstructed genome (MAGs) catalog. And we also recall that all in silico reconstructed genomes need a biological confirmation of the existence of the corresponding microbial entity. Please see the blue highlighted revision part of the introduction.

Comment 2: Line 40: what do the authors mean by “complex behaviors”?

Response: 1) Complex behaviors have been used in genetics to refer those behavior traits that are controlled by multi-genetic loci with additive and/or non-addition

genetic effects. As behaviors are genetically suggested being affected by multi-genetic loci, we used the word “complex behaviors”.

2) We thank for the reviewer’s remind, we revised the sentence as “which play vital roles in host metabolism, immunity, and even behaviors.”

Comment 3: Line 42: the authors wrote “Most of these studies focused on microbes with sequenced genomes”. Indeed, is it true? Most studies rely on available annotations that are often connected to partial genome sequences. Could the authors be more specific in this statement?

Response: We agree with this comment, and revised this sentence correspondingly.

Most of these studies relied on the available annotation information of microbes that are often connected to partial genome sequences, e.g. hypervariable regions of 16S rRNA gene.

Comment 4: Line 49: suggestion to replace “host phenotypes (diseases)” by “host phenotypes and diseases”. All associated phenotypes are not related to diseases.

Response: We appreciate this suggestion, and revised this sentence correspondingly.

Comment 5: Lines 52 -54: “Reference genomes of microbiota are essential resources for understanding the functional role of specific microbes in the microbiome and for quantifying their abundance in metagenomes”. Quite challenging to put in the same

sentence microbiota, microbiome and metagenomes. The authors might refer to Berg et al., 2020 for instance who have proposed to clarify the definition of each term (<https://pubmed.ncbi.nlm.nih.gov/32605663/>).

Response: We appreciate the reviewer for indicating this. We revised this sentence as follows: Reference genomes are essential resources for understanding the functional role of specific microbes and quantifying their abundance in the gut microbiome.

Comment 6: Again, since the first gene catalog for pigs was produced from only 287 fecal samples, it is not very appropriate to compare it with the integrated human catalog. The authors could instead promote the idea that their aim was to build an integrated catalog for the pig as it had been done in humans. The integrated catalog in humans added rare genes mostly.

Response: We agree with the reviewer's comment, and deleted the sentence "compared with the integrated gene catalog of the human gut microbiome, the PGC catalog was constructed from a relatively small sample size (1,267 vs. 287)" from the manuscript. We also added "to construct an integrated gene catalog and recover MAGs of the pig gut microbiome," to the manuscript.

Comment 7: Lines 75-88: this ending paragraph is more a long summary of the results than a presentation of the main objectives of the study. In addition, the relevance to combine data from many fecal DNA (472) with cecum lumen (20), ileal

lumen (6) and jejunum lumen (2) is not very clear. If the aim is to use a unique catalog to study any gut locations, that has to be stated in the aims of the study.

Response: We appreciate the reviewer for this suggestion, and made the revisions correspondingly.

In order to construct integrated an gene catalog and recover MAGs of the pig gut microbiome, we performed the present research to sequence five hundred samples from a wide range of sample sources spanning various ages, breeds, geographical areas, domestication and gut locations. Especially, the lumen samples from jejunum, ileal and cecum were used to improve the representation of this integrated gene catalog on the microbiome of whole intestinal tract. Furthermore, the dataset of the PGC catalog was also integrated into the construction of the catalog. We constructed gene catalogs (named pig integrated gene catalog, PIGC) of the pig gut microbiome consisting of 48,697,887 (PIGC100), 17,237,052 (PIGC90), and 7,246,447 (PIGC50) non-redundant genes at 100%, 90%, and 50% amino acid identity, respectively. In addition, a total of 6,339 MAGs were recovered, which were clustered to 2,673 species-level genome bins (SGBs), of which more than 86% (2,309) had no available genome sequence in the current database (unknown SGBs, uSGBs). To demonstrate the value of these resources, we used the catalogs of microbial genes and MAGs to compare the gut microbiomes of wild boars and commercial Duroc pigs, which represent pigs raised in two extremely different conditions (free-living *vs.* standard farm-raised in the pig industry) to identify the detailed microbiome differences

between these two cohorts.

Comment 8: 2) Results

Lines 134-144: was it not expected that the higher the sequencing depth the higher the gene number? Is that an original result that needs to be presented with so many details? That is interesting to report the limited number of added genes provided by young pigs (before weaning, limited microbial diversification), gut sites and wild boars. It would be interesting to know whether these specific genes are rare or in high abundance in the samples, they came from.

Response: We revised the manuscript accordingly based on the reviewer's comments.

The description about the result of the contribution of the sequencing depth to gene number were cut accordingly. Please see line 141-146, page 7. In addition, we analyzed the abundances of these sample source-specific genes in the samples that they came from. The result was added to the manuscript (line 164-172), and also listed in the following:

We further analyzed the abundances of these sample source-specific genes. Notably, high proportions of small intestine lumen-specific, piglet sample-specific and wild boar sample-specific genes had the abundances of \geq average abundance of gene set in the corresponding sample source. In particular, more than 65% of small intestine lumen-specific genes showed high abundance. However, most of feces sample-specific genes of domestic pigs (97%) showed low abundances in the samples

that they came from (Figure 2e). This result suggested that the utilization of the samples from different gut locations and wild boars provided very useful gene set to improve the representation of the PIGC catalog.

Figure. The proportions of sample source-specific genes having high abundance (\geq average abundance) in the corresponding samples that they came from.

Comment 9: If the reviewer understands well, the authors reported that the Duroc-JY pigs seem closer to the wild boars than to the Duroc-SH (cf. supp. Fig. 8). Are there hypotheses for this finding? Was it expected that the alpha diversity of the fecal DNA from wild boars was not higher than from the highly selected pigs?

Response: *P. copri* had an extremely high abundance in the gut of Duroc-SH. This reduced the α -diversity significantly. We indicated this in the manuscript. Please see line 274-277, page 13, and the following:

A significantly lower α -diversity was observed in Duroc-SH pigs, which might be caused by the extremely high abundance of *P. copri* in the gut of Duroc-SH (the

overgrowth of *P. copri*, 54.06% in relative abundance).

As for the comparison of the α -diversity of gut microbiome between wild boars and Duroc pigs, this comparison was performed at the species level, and we did not observe significant difference. We understand that this might be caused by the poor annotations of metagenomic sequencing data at the species level. We added an analysis at the genus level, and the wild boars showed a higher α -diversity of the gut microbiome although the difference was not achieved significance level between wild boars and Duroc-JY.

We added this result to the manuscript, and this paragraph was revised as the following:

Compared with Duroc pigs, wild boars had higher α -diversity of the gut microbiome at the genus level although the difference was not achieved significance level between wild boars and Duroc-JY (**Supplementary Fig. 9b**). However, this difference was not observed at the species level (**Supplementary Fig. 9c**). This should be caused by the poor annotations of metagenomic sequencing data at the species level. A significantly lower α -diversity was observed in Duroc-SH pigs, which may be the result of an extremely high abundance of *P. copri* in the gut of Duroc-SH (the overgrowth of *P. copri*, 54.06% in relative abundance).

Comment 10: Supp. Table 6: add the unit for the mean values (columns F, G, H)

Response: Following this comment, we added the unit of Fragments Per Kilobase of

gene sequence per Million mapped fragments (FPKM) for the mean values in the Supplementary Table 3, 6, 7 and 8. Please see Supplementary Tables.

Comment 11: Line 270: remove “reported in another paper under review”

Response: According to this comment, we deleted “reported in another paper under review” from this sentence.

Comment 12: Figure 1: explain all abbreviations and detail that the % identity is at the amino acid level.

Response: We revised the figure legend of Figure 1 accordingly. More details were added. Please see the following and the manuscript.

Figure 1 Pipeline for the construction of Pig Integrated Gene Catalog (PIGC) and metagenome-assembled genomes (MAGs). Metagenomic sequencing data from the samples spanning age, sex, breed, gut location, geography, and domestication, as well as a pig gene catalog (PGC) from 287 metagenome data were integrated and used to construct the PIGC catalog. The complete genes were clustered at 100%, 90% and 50% amino acid identity to generate non-redundant gene catalogs of PIGC100, PIGC90 and PIGC50. The reconstructed microbial genomes were clustered to strain-level and species-level genome bins (SGBs) at 99% and 95% of the average nucleotide identity (ANI), respectively. The 6,339 non-redundant MAGs were divided into medium-quality MAGs (more than 50% completeness and less than 5%

contamination) and high-quality MAGs (more than 90% completeness and less than 5% contamination). SGBs containing at least one reference genome (or metagenome-assembled genome) in the Genome Taxonomy Database (GTDB) were considered as known SGBs (kSGB). The SGBs without reference genomes were considered as unknown SGBs (uSGBs).

Comment 13: Line 817; replace pilglet by piglet

Response: Thank you for indicating this syntax error. We made the correction correspondingly.

Comment 14: Figure 3: *Prevotella* species were not found in the predominant core bacteria taxa in fecal DNA whereas they were identified as predominant in Duroc pigs for instance (e.g. *Prevotella copri*). Could this finding be discussed?

Response: *Prevotella copri* was a core bacteria species found in all three gut locations (ileum lumen, cecum lumen and feces). It had the highest abundance in feces samples of Duroc pigs among all tested samples (The relative abundance of *P. copri* achieved 23.38% and 54.06% in Duroc-JY and Duroc-SH, respectively). The high abundance of *P. copri* in the gut of Duroc pigs should be due to the feeding of commercial formula feed containing high levels of energy and protein.

Comment 15: Title of supplementary table 8: replace “Comparison of antibiotic resistance between wild boars and Duroc pig” by Comparison of antibiotic resistance gene prevalence between wild boars and Duroc pigs”

Response: Because the sample size was not large enough, we compared the abundance of antibiotic resistance genes rather than the prevalence between wild boars and Duroc pigs, so the title of Supplementary Table 8 was revised as “Comparison of the abundance of antibiotic resistance genes between wild boars and Duroc pigs”.

Comment 16: Discussion

It is often reported that intensive selection has led to a decrease in domestic animal biodiversity and that wild boars are more prone to efficiently face pathogens. Have the data provided any clues relating to these general statements? Comparison between Duroc pigs and wild boars could contribute to discuss this issue.

Response: As the response for comment 9, compared with Duroc pigs, wild boars had higher α -diversity of the gut microbiome at the genus level although the difference

was not achieved significance level between wild boars and Duroc-JY (Supplementary Fig. 9b).

As for the tendency that wild boars are more to efficiently face pathogens, we discussed this in the manuscript. Please see line 357-379, page 17-18. We also list this discussion in the following:

The gut microbiome of wild boars had a significantly higher abundance of *Bacteroides* spp. (Fig. 5a). *Bacteroides* play important roles not only in producing valuable nutrients and energy by breaking down food, but also in regulating immune abilities. For example, *B. uniformis* and *B. xylanisolvens* can utilize both dietary and endogenous glycans, along with the production of beneficial end products, such as short chain fatty acids (SCFAs), for both the bacterium and the host^{44,45}. *Bacteroides ovatus* is a representative of *Bacteroides* genus that has immune-regulatory abilities⁴⁶. *Bifidobacterium* spp. were also significantly enriched in the gut microbiome of wild boars. *Bifidobacteria* can utilize a diverse range of plant-derived oligo- and polysaccharides that can't be degraded by host⁴⁷ and produce SCFAs. SCFAs not only provide the energy for the survival of wild boars⁴⁸, but also play roles in anti-inflammation and improving immunity^{49,50}. The relative abundance of *P. copri* in Duroc-JY and Duroc-SH pigs achieved 23.38% and 54.06%, respectively (Supplementary Fig. 9d). In a previous study, we showed the causative role of *P. copri* in host fat accumulation that rely on formula diets and induces by chronic inflammation. Unlike wild boars, in the modern pig industry Duroc pigs are fed the

commercial formula feed containing high levels of energy (3,023kcal/kg for Duroc-SH and 2,960 kcal/Kg for Duroc-JY) and protein to exploit growth potential, which induces an overgrowth of *P. copri* in the gut (**Supplementary Fig. 9d**). *S. suis*, a pathogen⁵², was also enriched in Duroc pigs. Pigs raised on commercial farms with high density and fed commercial formula feed easily become sick.

All authors appreciate the editor and reviewers again for these invaluable comments and suggestion. We look forward to hearing from you at your earliest convenience. Your kind consideration for publication in *Nature Communications* will be greatly appreciated.

Please inform me if further information is required.

Yours sincerely

Prof. Lusheng Huang

President and Professor, Jiangxi Agricultural University
Member, Chinese Academy of Science (CAS)
Director, National Key Laboratory for Swine Genetic Improvement and Production Technology,
President, National Committee for Farm Animal Genetic Resources, Ministry of Agriculture and
Rural affairs of China
NanChang, 330045, P.R.China.
Tel: 0086 791 3813080,

Fax: 0086 791 3900189